# Flattened 1D fragments of fullerene C$_{60}$ that exhibit robustness toward multi-electron reduction

Masahiro Hayakawa [1,2,6], Naoyuki Sunayama[3,6], Shu I. Takagi [3], Yu Matsuo[3], Asuka Tamaki[2], Shigehiro Yamaguchi [2,4], Shu Seki [5] & Aiko Fukazawa [1] ✉

Fullerenes are compelling molecular materials owing to their exceptional robustness toward multi-electron reduction. Although scientists have attempted to address this feature by synthesizing various fragment molecules, the origin of this electron affinity remains unclear. Several structural factors have been suggested, including high symmetry, pyramidalized carbon atoms, and five-membered ring substructures. To elucidate the role of the five-membered ring substructures without the influence of high symmetry and pyramidalized carbon atoms, we herein report the synthesis and electron-accepting properties of oligo(biindenylidene)s, a flattened one-dimensional fragment of fullerene C$_{60}$. Electrochemical studies corroborated that oligo(biindenylidene)s can accept electrons up to equal to the number of five-membered rings in their main chains. Moreover, ultraviolet/visible/near-infrared absorption spectroscopy revealed that oligo(biindenylidene)s exhibit enhanced absorption covering the entire visible region relative to C$_{60}$. These results highlight the significance of the pentagonal substructure for attaining stability toward multi-electron reduction and provide a strategy for the molecular design of electron-accepting π-conjugated hydrocarbons even without electron-withdrawing groups.

Fullerenes have fascinated scientists in broad research fields since the discovery of buckminsterfullerene C$_{60}$[1]. In addition to the attractive spherical structure with high symmetry, their electron-accepting characteristic is notable among π-conjugated materials. Fullerenes have exceptional stability toward multi-electron reduction, unlike other electron-accepting π-conjugated systems[2,3]. For example, C$_{60}$ reportedly undergoes 6-electron and 12-electron reductions in solution[2] and the solid state, respectively[3]. This robustness toward multi-electron reduction enables access to various alkaline metal salts, among which Cs$_3$C$_{60}$ exhibits superconductivity with the transition temperature ($T_c$) of 38 K, which is the highest value among all the molecular materials[4–6]. Moreover, their moderately low-lying lowest unoccupied molecular orbital (LUMO)[7–9] and high electron mobility[10,11] have rendered them a central role as electron-transporting materials[12]. It is worth noting that this high electron affinity is realized based on the carbon-only framework. This is unlike the molecular design of most electron-accepting organic materials, which relies on introducing electron-withdrawing atoms or groups, such as fluorine[13–15], chlorine[14], cyano[16–18], carbonyl[17–19], and imine moieties[17,19], into π-conjugated systems.

[1]Institute for Integrated Cell-Material Sciences (WPI-iCeMS), Institute for Advanced Study, Kyoto University, Yoshida, Sakyo-ku, Kyoto 606-8501, Japan. [2]Department of Chemistry, Graduate School of Science, and Integrated Research Consortium on Chemical Science (IRCCS), Nagoya University, Furo, Chikusa, Nagoya 464-8602, Japan. [3]Department of Energy and Hydrocarbon Chemistry, Graduate School of Engineering, Kyoto University, Yoshida, Sakyo-ku, Kyoto 606-8501, Japan. [4]Institute of Transformative Bio-Molecules (WPI-ITbM), Nagoya University, Furo, Chikusa, Nagoya 464-8601, Japan. [5]Department of Molecular Engineering, Graduate School of Engineering, Kyoto University, Nishikyo-ku, Kyoto 615-8510, Japan. [6]These authors contributed equally: Masahiro Hayakawa, Naoyuki Sunayama. ✉e-mail: afukazawa@icems.kyoto-u.ac.jp

Hence, the minimum structural basis for the high electron affinity and exceptional stability toward the multi-electron reduction of fullerenes are of interest. Several structural factors have been proposed, including the degeneracy of LUMO and LUMO + 1 due to the highly symmetrical structure (Fig. 1a), the mitigation of bond angle strain around the carbon atoms upon reduction due to the inherently pyramidalized geometry (Fig. 1b), and the presence of five-membered ring substructures that can acquire Hückel aromaticity in a reduced state (Fig. 1c)[8,20,21]. Although π-conjugated hydrocarbons with fragment structures of fullerenes can be promising for understanding the effect of each factor, most fullerene fragment molecules reported to date, such as corannulene[22,23], sumanene[24], and larger molecules[25–30], have bowl-shaped structures in which most of the carbon atoms adopt pyramidalized geometries (Supplementary Fig. 1). This fact indicates that the curved structure of the fullerenes is of significant interest. Although Brunetti and coworkers reported π-extended 9,9′-bifluorenylidene derivatives composed of a $C_{60}$ substructure without having pyramidalized carbon atoms[31], their highly twisted structures impede the effective extension of the π-conjugation, limiting the contribution of five-membered rings. A fragment molecule of $C_{60}$ with effective π-conjugation between the five-membered ring substructures without pyramidalized carbon atoms is necessary to clarify the role of five-membered rings on the high electron affinity and robustness toward multi-electron reduction of fullerenes.

Therefore, to elucidate the role of the five-membered ring substructures in the exceptional electron affinity of fullerenes without the influence of high symmetry and pyramidalized geometries of carbon atoms, we conceived a molecular design of π-conjugated oligomers **3** and **4** composed of a one-dimensional fragment of $C_{60}$ in their main chains (Fig. 1e) based on the following idea. First, we focused on the hoop-shaped substructure of $C_{60}$ wherein 6 five-membered rings were linearly connected, that is, cyclic ter(1,1′-biindenylidene) **1** (Fig. 1d). Next, based on the similarity of the π-conjugated hoops and corresponding linear π-conjugated polymers, excluding their symmetry and structural distortion[32,33], we designed linear π-conjugated polymer **2** with an identical repeating unit to eliminate the pyramidalization effect. We also anticipated that it could be an excellent platform for demonstrating the effect of five-membered rings on the stability toward the multi-electron reduction as the number of five-membered rings per molecule can be increased by elongating the chain length. Poly(pentafulvene)s, including **2**, have previously been studied as candidate structures for narrow- or zero-bandgap polymers by quantum chemical calculations[34–37], although their electron-accepting characteristics have not yet been discussed. Moreover, the synthesis of π-conjugated oligomers or polymers of pentafulvenes has not yet been reported. In this study, to clarify the molecular structures and chain-length dependence of the properties, we designed **3** and **4**, wherein both ends of the one-dimensional $C_{60}$ fragment **2** were end-capped with phenyl and 4-silylphenyl groups, respectively (Fig. 1e).

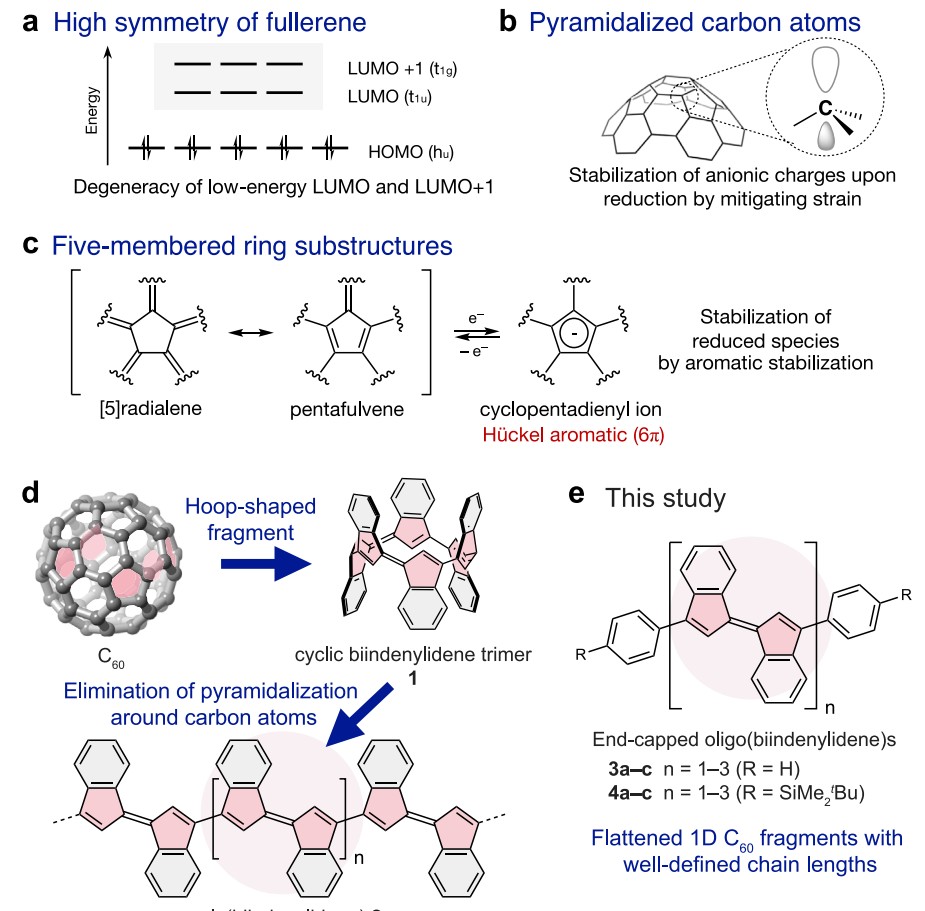

**a** High symmetry of fullerene

LUMO +1 ($t_{1g}$)
LUMO ($t_{1u}$)
HOMO ($h_u$)
Degeneracy of low-energy LUMO and LUMO+1

**b** Pyramidalized carbon atoms

Stabilization of anionic charges upon reduction by mitigating strain

**c** Five-membered ring substructures

[5]radialene — pentafulvene ⇌ cyclopentadienyl ion
Hückel aromatic (6π)

Stabilization of reduced species by aromatic stabilization

**d**

$C_{60}$
Hoop-shaped fragment
cyclic biindenylidene trimer **1**
Elimination of pyramidalization around carbon atoms
poly(biindenylidene) **2**

**e** This study

End-capped oligo(biindenylidene)s
**3a–c** n = 1–3 (R = H)
**4a–c** n = 1–3 (R = SiMe₂$^t$Bu)

Flattened 1D $C_{60}$ fragments with well-defined chain lengths

**Fig. 1 | Origin of prominent electron-accepting character of fullerenes.** Three proposed structural factors for the electron-accepting character of fullerenes: **a** high symmetry that results in the triply degenerated LUMO and LUMO + 1, **b** pyramidalized carbon atoms that mitigate bond angle strain around the carbon atoms upon reduction, and **c** five-membered ring substructures that can acquire aromatic character in a reduced state. **d** Our molecular design of π-conjugated polymer **2** based on the hoop-shaped substructure **1** of $C_{60}$. **e** End-capped oligo(-biindenylidene)s **3** and **4** examined in this study.

**Fig. 2 | Synthesis of biindenylidene oligomers. a** Synthesis of aryl-capped biindenylidene monomers **3a** and **4a** by the Suzuki−Miyaura coupling using dibromobiindenylidene **5**. **b**, **c** Iterative synthesis of biindenylidene dimers **3b** and **4b** and trimers **3c** and **4c** by the Miyaura-Ishiyama borylation and Suzuki−Miyaura coupling. B₂pin₂ = bis(pinacolato)diboron, KOAc = potassium acetate.

Herein, we report the syntheses and properties of **3** and **4** up to n = 3. X-ray crystallographic analyses showed that the $C_{60}$ fragment structures in **3** and **4** were unaffected by the pyramidalization of carbon atoms. Electrochemical and photophysical studies revealed the role of five-membered rings in the exceptional electron-accepting characteristic of $C_{60}$ without electron-withdrawing groups. Moreover, comparison of the electronic structures of **3** and **4** with those of $C_{60}$ revealed the characteristics of oligo(biindenylidene)s, reflecting one-dimensional π-conjugation between the five-membered rings.

## Results
### Synthesis and characterization
Oligo(1,1′-biindenylidene)s **3** and **4** were synthesized by iterative cross-coupling reactions using 3,3′-dibromo-1,1′-biindenylidene (**5**)[38,39] as a precursor. The Pd-catalyzed coupling reaction of **5** with phenylzinc chloride under typical Negishi-coupling conditions resulted in the formation of a complex mixture containing both the expected end-capped products, such as monophenylated biindenylidene **6a** and diphenylated biindenylidene **3a**, and the unexpected oligomers such as the monophenylated biindenylidene dimer **6b**, diphenylated biindenylidene dimer **3b**, and trimer **3c** (Supplementary Discussion 1 and Supplementary Fig. 2). Conversely, the Suzuki−Miyaura coupling using organoboronates successfully produced the end-capped biindenylidenes without the formation of the undesirable homo-coupling processes (Fig. 2). Specifically, the reactions of **5** with 2.4 equivalents of phenylboronic acid or 4-(t-butyldimethylsilyl)phenylboronate in the presence of $K_2CO_3$ and a catalytic amount of Pd(PPh₃)₄ gave the end-capped biindenylidenes **3a** and **4a** in 62% and 80% yields, respectively (Fig. 2a). Similarly, the Suzuki−Miyaura coupling of **5** with the reduced amount of aryl boronates afforded monoarylated bromobiindenylidenes **6a** and **7a**, which are useful precursors for the iterative synthesis of biindenylidene oligomers, in moderate yields without the formation

of homo-coupling products (Fig. 2b). The bromobiindenylidenes **6a** and **7a** were subjected to consecutive Miyaura-Ishiyama borylation and Suzuki−Miyaura coupling processes in one-pot in the presence of bis(pinacolato)diboron, Pd(PPh₃)₄, and potassium carbonate to afford the corresponding biindenylidene dimers **3b** (70% yield) and **4b** (55% yield), respectively (Fig. 2b). The trimers **3c** and **4c** were synthesized by the transformation of dibromobiindenylidene **5** to the corresponding diboronate **8** by the Miyaura-Ishiyama borylation (see also Supplementary Discussion 2), followed by the Suzuki−Miyaura coupling with 2.2 equivalents of monobromide **6a** or **7a**, respectively (Fig. 2c).

The molecular structures of oligo(biindenylidene)s **3a**−**c** and **4a**−**c** obtained were verified by nuclear magnetic resonance (NMR) spectroscopy and mass spectrometry, and those of the monomers (**3a** and **4a**) and dimers (**3b** and **4b**) were confirmed by single-crystal X-ray diffraction analyses (Fig. 3 and Supplementary Fig. 3, *vide infra*). Notably, **3a**−**c** and **4a**−**c** were stable under ambient conditions and could be handled without any precautions. Furthermore, a series of oligo(biindenylidene)s exhibited moderate solubility in common organic solvents. For example, the solubilities of **3b** and **3c** in $CH_2Cl_2$ were 5 and $2 \, g \, L^{-1}$, respectively, which are one order of magnitude higher than that of $C_{60}$ ($0.26 \, g \, L^{-1}$)[40].

Single-crystal X-ray diffraction analyses of **3b** and **4b** revealed one-dimensional π-conjugated frameworks wherein the biindenylidene units were directly bonded with the same connectivity as those in $C_{60}$ (Fig. 3a, b). Each biindenylidene unit had an *E* configuration, and all the adjacent units adopted *s*-trans conformation. The C−C bond lengths in the biindenylidene units of **3b** and **4b** were comparable to those of the corresponding **3a** and **4a** without any noticeable dimerization effect (Supplementary Table 2). Although their π-conjugated frameworks deviated from coplanarity, the conformation differed depending on the terminal substituents. In particular, the biindenylidene skeletons in phenyl-capped **3b** and trialkylsilylphenyl-capped **4b**

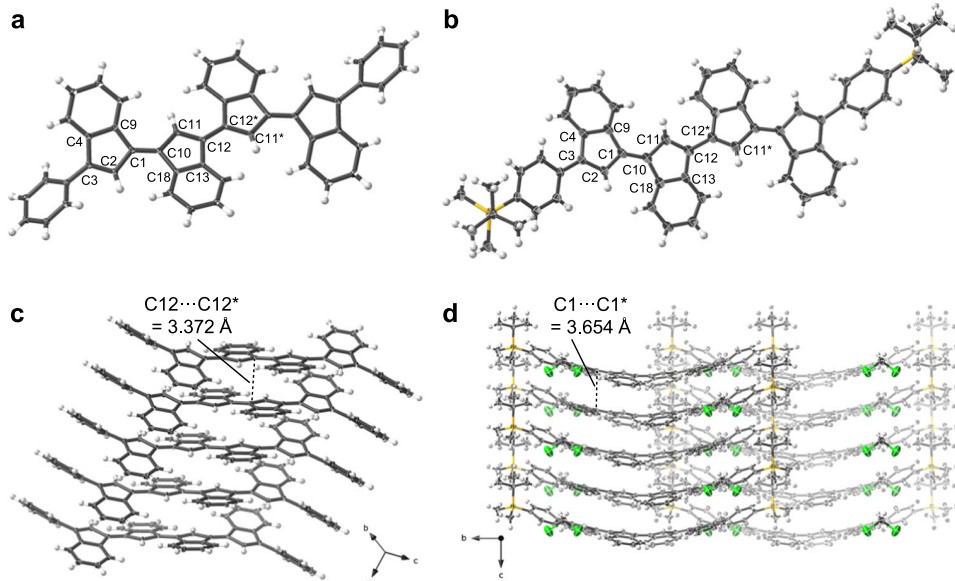

**Fig. 3 | X-ray crystal structures of biindenylidene dimers.** Molecular structures are drawn in thermal ellipsoid plots (50% probability for thermal ellipsoids; gray, carbon; white, hydrogen; yellow, silicon; green, chlorine). **a, b** Top view of **3b** and **4b**, respectively. CH$_2$Cl$_2$ molecules in the crystal lattices of **4b** are omitted for clarity. **c, d** Crystal-packing structure of **3b** and **4b**, respectively. The shortest intermolecular C⋯C distances were shown.

were slightly twisted to comparable extents, with intra-unit torsion angles (C2–C1–C10–C11) of 160.6° and −166.4°, respectively, because of the steric repulsion between peripheral hydrogen atoms. Conversely, the inter-unit torsion angles (C11–C12–C12*–C11*) were 180° and 152.8° for **3b** and **4b**, respectively, reflecting conformational flexibility around the freely rotatable C–C bonds (Fig. 3c, d). Consequently, the π-conjugated chains of **3b** and **4b** exhibited substantially different S-shaped conformations with a $C_i$ symmetry center and an arch-like shape with a $C_2$ symmetry axis, respectively. Accordingly, **3b** and **4b** adopted different packing motifs of the offset π-stacked arrays (Fig. 3c) and one-dimensional π-stacked columns (Fig. 3d), respectively. Reflecting the difference in packing motifs, the shortest interatomic C⋯C distance between stacked molecules in **3b** of 3.372 Å was significantly smaller than that in arched-shape **4b** (3.654 Å), indicative of more effective intermolecular interaction of π-orbitals.

Despite the marked difference in the conformation and the packing motifs, the torsional angles in **3b** and **4b** were sufficiently small for extending π-conjugation over the main chains. Despite these conformational differences, all carbon atoms in the biindenylidene skeletons of **3b** and **4b** maintained a planar geometry without pyramidalization. More specifically, the π-orbital axis vector (POAV) analysis[41] for the crystal structures of **3b** and **4b** revealed that the averaged values of the σ–π interorbital angle ($\theta_{\sigma\pi}$) for the carbon atoms that consist of five-membered rings were 90.7(6)° and 91.0(5)°, respectively (Supplementary Tables 4 and 6). These values are markedly smaller than those of C$_{60}$ (101.6°) and corannulene (98.2°), and comparable to that of the ideal sp$^2$ carbon atom like in graphite (90.0°)[42]. These results demonstrate that oligo(biindenylidene)s are suitable for studying the effect of five-membered rings on the electron affinity of π-conjugated hydrocarbons without the influence of the pyramidalization of carbon atoms.

### Electrochemical properties

To corroborate the dependence of the redox properties on the number of five-membered rings, the electrochemical properties of oligo(biindenylidene)s **3a–c** and **4a–c** were examined using cyclic voltammetry (Fig. 4a, Supplementary Figs. 5 and 6, and Supplementary Table 7). The cyclic voltammograms of trialkylsilylphenyl-capped **4a**, **4b**, and **4c** in tetrahydrofuran (THF) showed two-, four-, and five-step reversible redox processes in the reductive region, respectively (Fig. 4a), and irreversible redox processes in the oxidative region (Supplementary Fig. 6). Considering that the first redox wave of **4c** was characterized by two-electron redox processes based on peak current analyses (Supplementary Discussion 3), **4a**, **4b**, and **4c** underwent two-, four-, and six-electron reductions, respectively, within the electrochemical window of THF. The phenyl-capped oligomers **3a–c** also showed cyclic voltammograms essentially similar to those of **4a–c**, except that **3c** showed only four-step redox processes in the reductive region within the electrochemical window of THF (Supplementary Fig. 5). The absence of the fifth redox wave might be attributable to the low solubility of **3c** in THF (0.23 g L$^{-1}$), resulting in the fifth redox wave masked by the increase in baseline current due to the reduction of THF. Overall, these results demonstrate that oligo(biindenylidene)s can accept electrons up to equal to the number of five-membered rings in their main chains. Accordingly, the five-membered ring substructure ensures the prominent stability of the π-conjugated hydrocarbons toward multi-electron reduction, even without pyramidalized carbon atoms or electron-withdrawing substituents.

The number of five-membered rings also gave the significant impact on the electron affinity of oligo(biindenylidene)s. Similar to those of various π-conjugated oligomers, the half-wave potentials of the first redox processes in the reductive region ($E_{1/2,\text{red}1}$) of **4a–c** shifted in the positive direction as the chain length increased (**4a**: −1.48 V, **4b**: −1.19 V, and **4c**: −1.09 V versus ferrocene/ferrocenium (Fc/Fc$^+$)), reflecting the decrease in the LUMO energy levels via the effective interaction between the π* orbitals of the biindenylidene moiety. Accordingly, the $E_{1/2,\text{red}1}$ value of **4c** was close to that of C$_{60}$ (−0.89 V versus Fc/Fc$^+$), albeit slightly more negative. This result suggests that the oligo(biindenylidene) substructure of C$_{60}$ also plays a crucial role in its high electron affinity.

Oligo(biindenylidenes)s exhibit high electron affinity and stability toward multi-electron reduction close to fullerenes. However, electrochemical studies have also highlighted a feature of oligo(biindenylidene)s based on one-dimensional π-conjugated chains. Unlike C$_{60}$ that undergoes multi-electron reduction in a stepwise manner with almost constant peak separations, oligo(biindenylidene)s **4** showed distinct behavior depending on the chain length. In particular, the separation of the first and second reductive waves decreased from

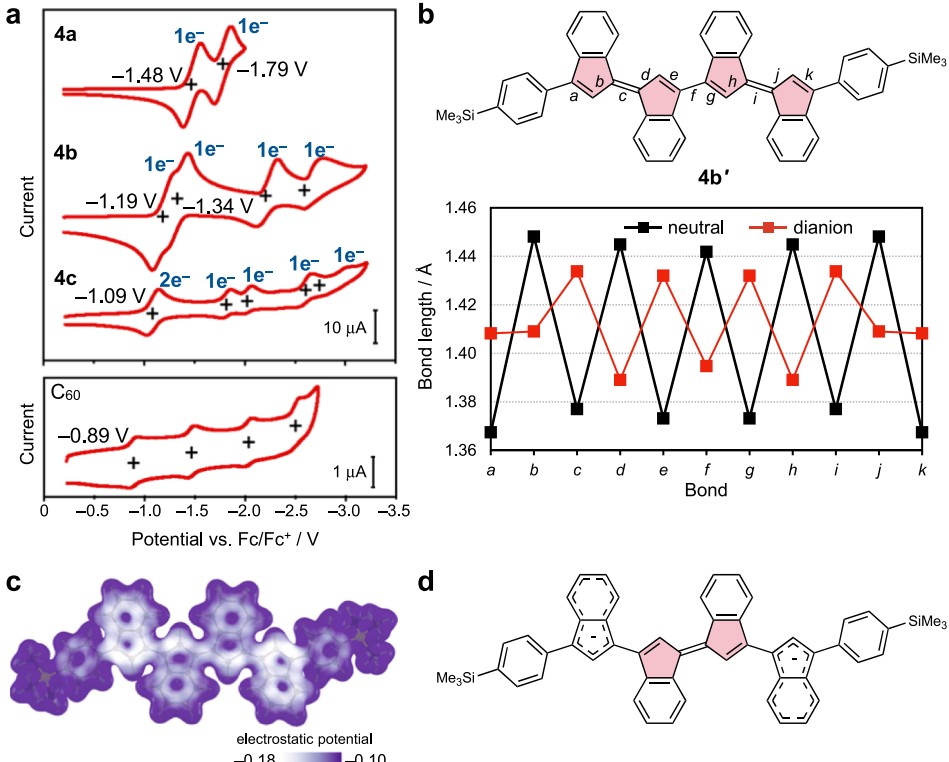

**Fig. 4 | Electrochemical properties of oligo(biindenylidene)s compared to those of $C_{60}$. a** Cyclic voltammograms of **4a**–**c** and $C_{60}$ measured at a scan rate of 0.1 V s$^{-1}$ in tetrahydrofuran using [$n$-Bu$_4$N][PF$_6$] (0.1 M) as the supporting electrolyte. All potentials are referenced against the ferrocene/ferrocenium (Fc/Fc$^+$) couple. **b** Plot of selected bond lengths in the optimized geometries of **4b′** in the charge-neutral (black) and dianionic states (red) calculated at the PBE0/6-31+G(d) level of theory. **c** Electrostatic potential map of [**4b′**]$^{2-}$ calculated at the MP2/6-31+G(d) level of theory. **d** The proposed structure of [**4b′**]$^{2-}$ based on the optimized geometry and charge distribution. Source data are available as a Source data file.

0.31 V (**4a**) to 0.15 V (**4b**) as the chain length increased, and these two waves merged into a single two-electron peak at **4c**. This peak separation decrease was in contrast with those of the subsequent reductive waves in **4b** and **4c**. The quantum chemical calculations of **4b′**, a model compound of **4b** wherein $t$-BuMe$_2$Si groups were simplified to Me$_3$Si groups to reduce the computational costs, demonstrated the localization of charge densities in the dianionic dimer ([**4b′**]$^{2-}$) on the indenylidene skeletons at both terminals (Fig. 4c). Moreover, the bond-length equalization as shown in Fig. 4b and Supplementary Fig. 11b as well as the negative values of the nuclear-independent chemical shifts (NICS, Supplementary Fig. 13) for the terminal indenylidene moieties in [**4b′**]$^{2-}$ strongly indicate the 10π Hückel aromaticity of the two indenyl anion moieties (Fig. 4d). Given these computational results, the localization of electron densities at both termini should be ascribed to the minimization of the electronic repulsion and the stabilization caused by the aromaticity of the two indenyl anion moieties. This charge separation in the dianion may be responsible for decreasing the peak separation of the first and second reduction waves as the chain length increases and reflects the structural feature of one-dimensional π-conjugated chains (see also Supplementary Discussion 5 for more detailed discussions including the results of radical anions [**4b′**]$^{2-}$).

## Electronic absorption

The ultraviolet/visible/near-infrared absorption spectra of **3** and **4** in CH$_2$Cl$_2$ were compared with that of $C_{60}$, highlighting the characteristics of oligo(biindenylidene)s (Fig. 5a and Supplementary Fig. 7). Similar to conventional π-conjugated oligomers, the increase in chain lengths of oligo(biindenylidene)s from **3a** to **3c** and from **4a** to **4c** resulted in a substantial redshift and an increase in the molar absorption coefficients ($\varepsilon$) of the longest-wavelength absorption

bands. In particular, the longest-wavelength absorption band of phenyl-capped **3c** with an absorption maximum wavelength ($\lambda_{max}$) of 653 nm was substantially red-shifted by 171 nm (5400 cm$^{-1}$) compared to that of the corresponding **3a** ($\lambda_{max}$ = 482 nm), and its $\varepsilon$ value of 5.34 × 10$^4$ M$^{-1}$ cm$^{-1}$ also increased (**3a**: 1.32 × 10$^4$ M$^{-1}$ cm$^{-1}$). Notably, all oligo(biindenylidene)s showed intense absorption bands with $\varepsilon$ values larger than 10$^4$ M$^{-1}$ cm$^{-1}$ in the visible region, which was in contrast to the weak absorption of $C_{60}$ ($\varepsilon$ < 10$^3$ M$^{-1}$ cm$^{-1}$) because of symmetry-forbidden transitions. In particular, the absorption spectra of **3c** and **4c** cover the entire visible region and reach the NIR region. This pronounced light absorption of oligo(biindenylidene)s might represent a characteristic property owing to their topological difference from $C_{60}$. Conversely, **3a**–**c** and **4a**–**c** were virtually non-luminescent in a degassed 2-methyltetrahydrofuran solution even at 77 K despite the allowed nature of the S$_0$ → S$_1$ transitions. Although the reason for this remains unclear, nonradiative decay processes via intersystem crossing or conical intersection may be responsible for the lack of fluorescence.

Time-dependent density functional theory (TD-DFT) calculations at the PBE0/6-31+G(d) level indicated that the highest-occupied molecular orbitals (HOMOs) and LUMOs of **3a**–**c** were characterized as delocalized π and π* orbitals over the main chains with a negligible contribution from fused and terminal benzene rings, supporting the effective orbital interactions between the biindenylidene units (Fig. 5b and Supplementary Fig. 9). In contrast to the low-lying LUMO levels of **3a**–**c** close to $C_{60}$, the HOMO levels of **3a**, **3b**, and **3c** (−5.76, −5.53, and −5.45 eV, respectively) were substantially higher than that of $C_{60}$ (−6.63 eV). These high-lying HOMO levels of **3c** are possibly responsible for the significantly narrower HOMO−LUMO gap than that of $C_{60}$, resulting a smaller S$_0$ → S$_1$ vertical excitation energy (**3c**: 1.72 eV, $C_{60}$: 2.16 eV). Furthermore, the oscillator strength $f$ of the S$_0$ → S$_1$ transition

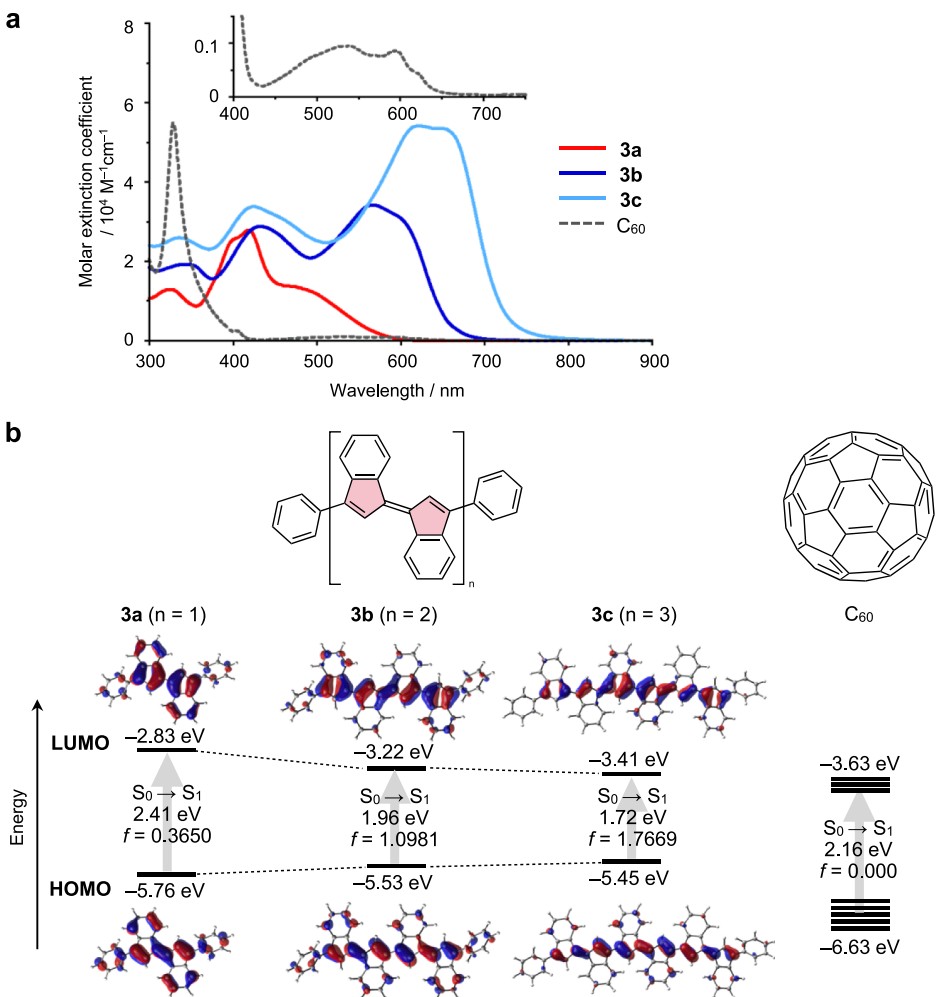

**Fig. 5 | Photophysical properties of oligo(biindenylidene)s compared with that of C$_{60}$. a** Ultraviolet/visible/near-infrared electronic absorption spectra of **3a** (red solid line), **3b** (blue solid line), **3c** (light-blue solid line), and C$_{60}$ (black broken line) in CH$_2$Cl$_2$. Inset: magnified spectrum in the wavelength range from 400−750 nm to clarify the absorption bands of C$_{60}$. **b** Energy diagrams and pictorial representations of the Kohn−Sham frontier molecular orbitals for the optimized geometries of **3a**−**c** (C$_1$ symmetry) and C$_{60}$ (I$_h$ symmetry) and the time-dependent DFT vertical excitations for the lowest-energy transitions calculated at the PBE0/6-31+G(d) level of theory. $f$ represents the oscillator strength of the lowest-energy transitions. Source data are available as a Source data file.

in **3c** ($f$ = 1.767) was larger than that of C$_{60}$ ($f$ = 0.000) because of its lower symmetry (Fig. 5b and Supplementary Table 12). These relatively high-lying HOMO and the symmetry-allowed characteristic of the S$_0$ → S$_1$ transition in the oligo(biindenylidene)s can be attributed to polyene-like one-dimensional π-conjugation.

### Mobility of electrons
To gain insights into the electron mobility of oligo(biindenylidene)s, the electronic photoconduction in a microcrystalline state was examined using flash-photolysis time-resolved microwave conductivity (FP-TRMC)[43,44]. Photoconductivity transients were observed for **3a**−**c** upon electrodeless photocarrier injection by the excitation at 355 nm (Fig. 6a and Supplementary Fig. 15). For **3a** and **3b**, almost no transient conductivity is observed whenever the excitation intensity is increased, or the crystal orientation is optimized. In stark contrast, **3c** exhibited fast decay with the transient conductivity $\phi\Sigma\mu$ ($\phi$: photogeneration efficiency of the charge carriers, $\Sigma\mu$: sum of the isotropic electron and hole mobility) more than one order of magnitude larger than those of **3a** and **3b** (Fig. 6a), indicating that trimer **3c** exhibits superior charge carrier conductivity compared to the shorter-chain oligomers. The major contribution to the

photoconductivity was determined to be electrons, given that the transient absorption spectrum of **3c** is in good agreement with the absorption band of radical anion **3c**$^{\cdot-}$ generated by electrochemical reduction (Fig. 6b and Supplementary Fig. 16). The photoinjected charge carrier density of **3c** was estimated by monitoring the transient absorption at 800 nm with absorption coefficients $\varepsilon$ of $7 \times 10^4$ M$^{-1}$ cm$^{-1}$ charge$^{-1}$ upon excitation at 355 nm, where photocarrier efficiency $\phi$ of $2 \times 10^{-3}$ and intrinsic intramolecular electron mobility $\mu_e$ of 0.06 cm$^2$ V$^{-1}$ s$^{-1}$ were obtained (Fig. 6c). Notably, the transient conductivities of **3c** and **4c** are comparable to each other despite the distinct crystal-packing structures and π−π stacking distances (Figs. 6d, 3c, and 3d), indicative of the predominant contribution of intramolecular electron transport for their electronic photoconduction. These results demonstrated a promising utility of long-chain oligo(biindenylidene)s as electron-transporting molecular wires, although inter-chain electron transport is negligible in the present system.

### Discussion
To elucidate the origin of the high electron affinity and stability toward the multi-electron reduction of fullerenes with only a carbon

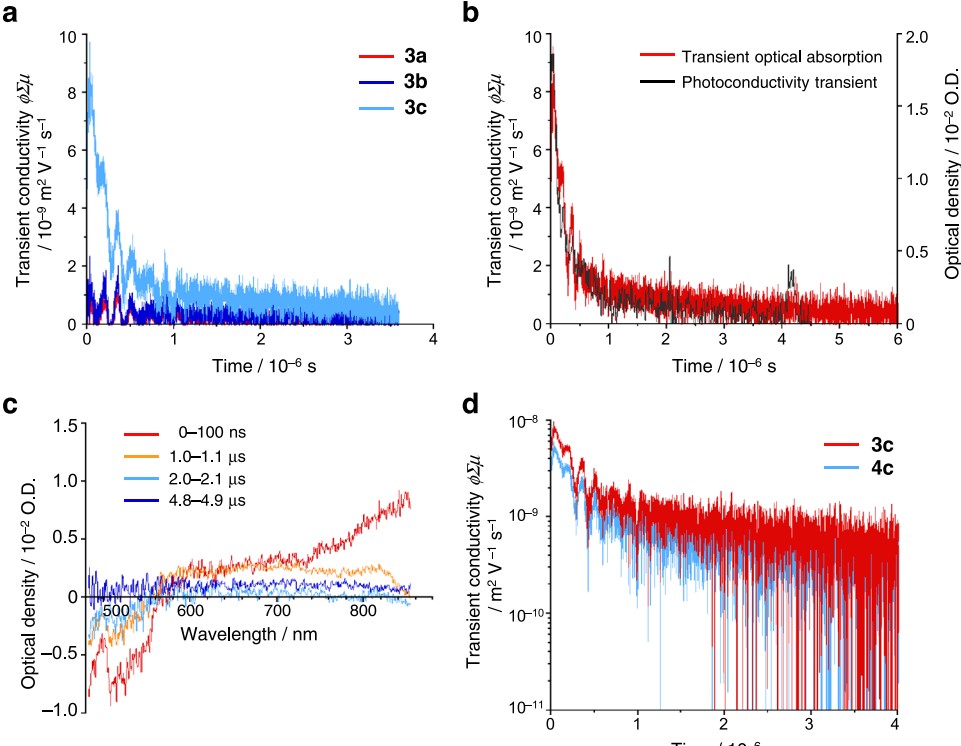

**Fig. 6 | Electronic photoconduction in a microcrystalline state evaluated by flash-photolysis time-resolved microwave conductivity (FP-TRMC) measurements. a** Kinetic traces of photoconductivity transients recorded for polycrystalline **3a** (red), **3b** (blue), and **3c** (light blue) under excitation at 355 nm, $4.6 \times 10^{15}$ photons cm$^{-2}$. **b** Comparison of a photoconductivity transient (black) and transient optical absorption (red) at 800 nm recorded for **3c**. Given the absorption coefficient of **3c**$^{·-}$ of ca. $7 \times 10^4$ M$^{-1}$ cm$^{-1}$, the values of $\phi$ and $\mu$ of **3c** are calculated to be $2 \times 10^{-3}$ and 0.06 cm$^2$ V$^{-1}$ s$^{-1}$, respectively. **c** Transient absorption spectra observed for polycrystalline **3c** under excitation at 355 nm, $3.6 \times 10^{16}$ photons cm$^{-2}$. Spectra were recorded at 0–100 ns (red), 1–1.1 μs (orange), 2–2.1 μs (light blue), and 4.8–4.9 μs (blue) after pulse exposure. **d** The photoconductivity transients recorded for **3c** (red) and **4c** (light blue) in the logarithmic scale, suggesting non-pseudo first-order recombination kinetics for the decay profiles. Source data are available as a Source data file.

skeleton, we designed and synthesized end-capped **3** and **4**, which are π-conjugated hydrocarbons composed of a one-dimensional fragment of buckminsterfullerene C$_{60}$ with the same connectivity between the five-membered rings. Crystallographic analysis of **3** and **4** confirmed that all sp$^2$ carbon atoms in their π-conjugated frameworks adopt trigonal planar geometries, unlike the pyramidalized carbon atoms in conventional fullerene fragment molecules. These oligo(biindenylidene)s have one-dimensional π-conjugated chains wherein the five-membered rings are directly connected. Electrochemical studies of oligo(biindenylidene)s **4** revealed that these oligo(biindenylidene)s can accept electrons up to equal to the number of five-membered rings in their main chains and experimentally corroborated that the five-membered ring substructures play a crucial role in attaining robustness toward multi-electron reduction. Notably, ter(biindenylidene)s, which are small-molecule π-conjugated hydrocarbons that do not have a rigid structure or curvature like fullerenes, can accept up to six electrons without noticeable decomposition.

In contrast to the similarity with fullerenes in terms of electron affinity, the one-dimensional π-conjugation in oligo(biindenylidene)s resulted in pronounced absorption covering the entire visible region, unlike the weak absorption of C$_{60}$ attributable to the highly symmetrical structure. Furthermore, FP-TRMC measurements indicated that the oligo(biindenylidene)s with longer chain lengths serve as promising electron-transporting molecular wires. The current results highlight the significance of the pentagonal substructure for attaining stability toward multi-electron reduction and provide a strategy for the molecular design of electron-accepting π-conjugated hydrocarbons even without electron-withdrawing groups.

## Methods

### Materials and characterization

The names of the molecules in the manuscript do not necessarily follow IUPAC recommendations to guide the reader. Melting points (mp) were determined with a Yanaco MP-S3 instrument. $^1$H and $^{13}$C{$^1$H} NMR spectra were recorded with a JEOL ECA 400 II spectrometer (100 MHz for $^{13}$C), a Bruker AVANCE III spectrometer (500 MHz for $^1$H and 125 MHz for $^{13}$C), or a JEOL JNM-ECZ600R spectrometer (600 MHz for $^1$H and 150 MHz for $^{13}$C) in chloroform-$d$ (CDCl$_3$), dichloromethane-$d_2$ (CD$_2$Cl$_2$), or 1,1,2,2-tetrachloroethane-$d_2$ (TCE-$d_2$). The chemical shifts in $^1$H NMR spectra are reported in $\delta$ ppm. using the residual proton of the solvents, i.e., CHCl$_3$ (7.26 ppm), CH$_2$Cl$_2$ (5.32 ppm), and 1,1,2,2-tetrachloroethane (6.00 ppm) as an internal standard, and those in $^{13}$C{$^1$H} NMR spectra are reported in $\delta$ ppm using the solvent signals of CDCl$_3$ (77.16 ppm), CD$_2$Cl$_2$ (53.84 ppm), and TCE-$d_2$ (73.78 ppm) as an internal standard. High-Resolution mass spectra (HRMS) were measured with a Bruker solarix (FT-ICR) system with the ionization method of APCI or MALDI. Thin-layer chromatography (TLC) was performed on plates coated with 0.25 mm thickness of silica gel 60F$_{254}$ (Merck). Column chromatography was performed using silica gel Wakosil® HC-N (FUJIFILM Wako Chemicals). Recycling preparative gel permeation chromatography (GPC) was performed using LaboACE LC-5060 (Japan Analytical Industry) equipped with a polystyrene gel column (JAIGEL-2HR-40, Japan Analytical Industry) using CHCl$_3$ as an eluent, or LC-Forte/R (YMC) equipped with a polystyrene gel column (YMC-GPC T-2000 and T-4000, YMC) using CH$_2$Cl$_2$ as an eluent. Anhydrous THF was purchased from Kanto Chemicals and further purified by Glass Contour solvent purifier systems. Bromobenzene and potassium acetate (KOAc) were purchased from Nacalai Tesque, Pd(PPh$_3$)$_4$,

bis(pinacolato)diboron, and $K_2CO_3$ from FUJIFILM Wako Chemicals, $ZnCl_2$(tmeda) from Sigma-Aldrich, for reagent grade and used without further purification. (E)-3,3′-Dibromo-1,1′-biindenylidene (5)[39] and 1-bromo-4-(tert-butyldimethylsilyl)benzene[45] were prepared according to the literature methods.

## Synthesis

Selected procedures are shown below. The characterization data are described in Supplementary Data 1 in the Supplementary Information.

### Synthesis of oligo(biindenylidene) 3a–c and 6a–b by the Negishi coupling of 5 with PhZnCl

To a solution of bromobenzene (335 mg, 2.13 mmol) in anhydrous THF (10 mL) was added a solution of n-BuLi in hexane (1.6 M, 1.45 mL, 2.32 mmol) dropwise over 2 min at −78 °C. After stirring for 1 h, $ZnCl_2$(tmeda) (593 mg, 2.35 mmol) was added to this solution, and the resulting mixture was stirred for 15 min at the same temperature to give a solution of PhZnCl. This solution was added dropwise to a solution of 5 (1.16 g, 3.00 mmol) and Pd(PPh$_3$)$_4$ (104 mg, 0.090 mmol) in anhydrous THF (3 mL) at −78 °C. After stirring for 21 h at 70 °C, the resulting mixture was quenched by $H_2O$, diluted with $CH_2Cl_2$, and separated into two layers. The aqueous layer was extracted with $CH_2Cl_2$ (3 × 50 mL). The combined organic layer was washed with brine and dried over anhydrous $Na_2SO_4$. After filtration and removal of all volatiles under reduced pressure, the crude product was subjected to silica gel column chromatography (hexane to $CH_2Cl_2$, then $CH_2Cl_2$ to ethyl acetate as eluent, $R_f = 0.78$ (hexane/ $CH_2Cl_2$ 1/1)), and further purified by recycling preparative GPC (CHCl$_3$ as eluent) to afford 40 mg of 3a as red solids (0.10 mmol, 4% yield), 12 mg of 3b as black solids (0.020 mmol, 1% yield), 7.3 mg of 3c as dark green solids (8.8 μmol, 0.9% yield), 169 mg of 6a as red solids (0.44 mmol, 15% yield), and 13 mg of 6b as black solids (0.021 mmol, 1% yield).

### Synthesis of oligo(biindenylidene)s 4a–c and 7a–b by the Negishi coupling of 5 with 4-(t-BuMe₂Si)C₆H₅ZnCl

This reaction was conducted in a similar manner as described for the reaction of 5 with PhZnCl. Thus, a solution of 4-(tert-butyldimethylsilyl) phenylzinc chloride was prepared by the halogen-lithium exchange of 1-bromo-4-(tert-butyldimethylsilyl)benzene (571 mg, 2.11 mmol) in anhydrous THF (10 mL) using n-BuLi in hexane (1.6 M, 1.45 mL, 2.32 mmol) followed by the transmetalation with $ZnCl_2$(tmeda) (584 mg, 2.31 mmol). The resulting solution of arylzinc chloride was subjected to the Negishi coupling with 5 (1.16 g, 3.00 mmol) in the presence of Pd(PPh$_3$)$_4$ (104 mg, 0.090 mmol) in anhydrous THF (4 mL). After standard aqueous workup and the extraction with $CH_2Cl_2$, the resulting crude product was subjected to silica gel column chromatography (hexane to $CH_2Cl_2$, then to ethyl acetate as eluent, $R_f = 0.22$ (hexane/$CH_2Cl_2$ 4/1)) and further purified by using recycling preparative GPC ($CH_2Cl_2$ as eluent) to afford 137 mg of 4a as red solids (0.23 mmol, 7% yield), 39 mg of 4b as black solids (0.047 mmol, 3% yield), 10 mg of 4c as dark green solids (9.4 μmol, 0.9% yield), 244 mg of 7a as red solids (0.49 mmol, 16% yield), and 10 mg of 7b as black solids (0.014 mmol, 0.9% yield).

### Synthesis of 1-(tert-butyldimethylsilyl)-4-(4,4,5,5-tetramethyl-1,3,2-dioxaborolan-2-yl)benzene

A solution of 1-bromo-4-(tert-butyldimethylsilyl)benzene (825 mg, 3.04 mmol), bis(pinacol)boronic acid (853 mg, 3.36 mmol), KOAc (901 mg, 9.18 mmol), and PdCl$_2$(dppf) (118 mg, 0.16 mmol) in 1,4-dioxane (30 mL) was stirred at 110 °C for 12 h. The resulting mixture was filtrated through a plug of Celite®, rinsed with THF, and the filtrate was concentrated under reduced pressure. The resulting crude mixture was suspended in n-hexane, filtrated through a plug of

Celite®, and rinsed with n-hexane. After the removal of all volatiles in vacuo from the resulting filtrate, the resulting crude product was purified by silica gel column chromatography (hexane to hexane/ ethyl acetate 95/5 as an eluent, $R_f = 0.41$ (hexane/ethyl acetate 9/1)) to afford 736 mg of the title compound as white solids (2.31 mmol, 76% yield).

### Improved synthesis of (E)-3,3′-diphenyl-1,1′-biindenylidene (3a) by Suzuki−Miyaura coupling

A solution of 5 (39 mg, 0.10 mmol), phenylboronic acid (30 mg, 0.24 mmol), $K_2CO_3$ (69 mg, 0.50 mmol), and Pd(PPh$_3$)$_4$ (2 mg, 2 μmol) in THF (2 mL) and $H_2O$ (0.4 mL) was stirred at 75 °C for 36 h. The resulting mixture was diluted with $CH_2Cl_2$, separated into two layers, and the aqueous layer was extracted with $CH_2Cl_2$ (3 × 15 mL). The combined organic layer was washed with brine, dried over anhydrous $Na_2SO_4$, and concentrated in vacuo. The resulting crude product was purified by silica gel column chromatography (hexane/$CH_2Cl_2$ 4/1 as an eluent, $R_f = 0.20$), followed by the further purification by recycling preparative GPC (CHCl$_3$ as an eluent) to afford 24.0 mg of 3a as red solids (0.062 mmol, 62% yield).

### Improved synthesis of (E)-3,3′-bis[4-(tert-butyldimethylsilyl) phenyl]-1,1′-biindenylidene (4a) by Suzuki−Miyaura coupling

A solution of 5 (38 mg, 0.099 mmol), 1-(tert-butyldimethylsilyl) 4-(4,4,5,5-tetramethyl-1,3,2-dioxaborolan-2-yl)benzene (77 mg, 0.24 mmol), $K_2CO_3$ (69 mg, 0.50 mmol), and Pd(PPh$_3$)$_4$ (3 mg, 2 μmol) in a mixed solvent of THF (2 mL) and $H_2O$ (0.4 mL) was stirred at 75 °C for 36 h. The resulting mixture was diluted with $CH_2Cl_2$, separated into two layers, and the aqueous layer was extracted with $CH_2Cl_2$ (3 × 20 mL). The combined organic layer was washed with brine, dried over anhydrous $Na_2SO_4$, and concentrated in vacuo. The resulting crude product was purified by silica gel column chromatography (hexane to $CH_2Cl_2$ as an eluent, $R_f = 0.96$ ($CH_2Cl_2$)), followed by the further purification by recycling preparative GPC (CHCl$_3$ as an eluent) to afford 48 mg of 4a as red solids (0.079 mmol, 80% yield).

### Improved synthesis of (E)-3-bromo-3′-phenyl-1,1′-biindenylidene (6a) by Suzuki−Miyaura coupling

A solution of 5 (1.93 g, 5.01 mmol), phenylboronic acid pinacol ester (1.03 g, 5.04 mmol), $K_2CO_3$ (3.47 g, 25.1 mmol), and Pd(PPh$_3$)$_4$ (291 mg, 0.252 mmol) in a mixed solvent of THF (100 mL) and $H_2O$ (20 mL) was stirred for 17 h at 75 °C in the dark. The resulting mixture was quenched by $H_2O$, diluted with $CH_2Cl_2$, and separated into two layers. The aqueous layer was extracted with $CH_2Cl_2$ (3 × 100 mL), and the combined organic layer was dried over anhydrous $Na_2SO_4$. After filtration and removal of all volatiles under reduced pressure, the crude product was purified by silica gel column chromatography (hexane only to hexane/ $CH_2Cl_2$ 1/1 as eluent, $R_f = 0.53$ (hexane/$CH_2Cl_2$ 4/1)) to afford 883 mg of 6a as red solids (2.30 mmol, 46% yield).

### Improved synthesis of (E)-3-bromo-3′-[4-(tert-butyldimethylsilyl)phenyl]-1,1′-biindenylidene (7a) by Suzuki−Miyaura coupling

A solution of 5 (1.93 g, 5.00 mmol), 1-(tert-butyldimethylsilyl)-4-(4,4,5,5-tetramethyl-1,3,2-dioxaborolan-2-yl)benzene (1.59 g, 5.00 mmol), $K_2CO_3$ (3.46 g, 25.0 mmol), and Pd(PPh$_3$)$_4$ (290 mg, 0.251 mmol) in a mixed solvent of THF (100 mL) and $H_2O$ (20 mL) was stirred for 17 h at 75 °C in the dark. The resulting mixture was quenched by $H_2O$, diluted with $CH_2Cl_2$, and separated into two layers. The aqueous layer was extracted with $CH_2Cl_2$ (3 × 100 mL), and the combined organic layer was dried over anhydrous $Na_2SO_4$. After filtration and removal of all volatiles under reduced pressure, the crude product was purified by silica gel column chromatography (hexane only to hexane/$CH_2Cl_2$ 19/1 as eluent, $R_f = 0.22$ (hexane)) to afford 1.18 g of 7a as red solids (2.37 mmol, 47% yield).

## Improved synthesis of 3,3‴-diphenylquaterindene (3b) by Suzuki−Miyaura coupling

A solution of **6a** (77 mg, 0.20 mmol), bis(pinacolato)diboron (26 mg, 0.10 mmol), $K_2CO_3$ (138 mg, 1.00 mmol), and Pd(PPh$_3$)$_4$ (12 mg, 10 μmol) in a mixed solvent of THF (2.0 mL) and $H_2O$ (0.4 mL) was stirred for 24 h at 75 °C in the dark. The resulting mixture was quenched by $H_2O$, diluted with $CH_2Cl_2$, and separated into two layers. The aqueous layer was extracted with $CH_2Cl_2$ (3 × 30 mL), and the combined organic layer was dried over anhydrous $Na_2SO_4$. After filtration and removal of all volatiles under reduced pressure, the crude product was washed with hexane to give 42 mg of **3b** as black solids (70 μmol, 70% yield).

## Improved synthesis of 3,3‴-bis[4-(*tert*-butyldimethylsilyl)phenyl]quaterindene (4b) by Suzuki−Miyaura coupling

A solution of **7a** (100 mg, 0.20 mmol), bis(pinacolato)diboron (25 mg, 0.10 mmol), $K_2CO_3$ (138 mg, 1.00 mmol), and Pd(PPh$_3$)$_4$ (12 mg, 10 μmol) in a mixed solvent of THF (2.0 mL) and $H_2O$ (0.4 mL) was stirred for 24 h at 75 °C in the dark. The resulting mixture was quenched by $H_2O$, diluted with $CH_2Cl_2$, and separated into two layers. The aqueous layer was extracted with ethyl acetate (3 × 20 mL), and the combined organic layer was dried over anhydrous $Na_2SO_4$. After filtration and removal of all volatiles under reduced pressure, the crude product was washed with hexane to give 46 mg of **4b** as black solids (55 μmol, 55% yield).

## Synthesis of (*E*)-3,3′-bis(4,4,5,5-tetramethyl-1,3,2-dioxaborolan-2-yl)-1,1′-biindenylidene (8)

A solution of **5** (79 mg, 0.20 mmol), bis(pinacolato)diboron (128 mg, 0.50 mmol), KOAc (105 mg, 1.07 mmol), and Pd(PPh$_3$)$_4$ (5 mg, 4 μmol) in THF (4 mL) was stirred at 75 °C for 12 h. The resulting mixture was quenched by $H_2O$, diluted by $CH_2Cl_2$, and separated into two layers. The aqueous layer was extracted with $CH_2Cl_2$ (3 × 15 mL). The combined organic layer was washed with brine and dried over anhydrous $Na_2SO_4$. After filtration and removal of all volatiles under reduced pressure, the crude product was purified by silica gel column chromatography (hexane/$CH_2Cl_2$ 4/1 to 1/1 as eluent, $R_f$ = 0.43 (hexane/$CH_2Cl_2$ 1/1)) to afford 39 mg of **8** as red solids (0.080 mmol, 40% yield).

## Improved synthesis of 3,3‴″-diphenylsexiindene (3c) by Suzuki−Miyaura coupling

A solution of **8** (48 mg, 0.10 mmol), **6a** (85 mg, 0.22 mmol), $K_2CO_3$ (70 mg, 0.51 mmol), and Pd(PPh$_3$)$_4$ (6.0 mg, 5.2 μmol) in THF (2.0 mL) and $H_2O$ (0.4 mL) was stirred for 12 h at 75 °C in the dark. After cooling the reaction mixture to an ambient temperature, the resulting precipitate was collected by vacuum filtration and washed with $H_2O$ and ethyl acetate to give 49 mg of **3c** as dark green solids (59 μmol, 59% yield).

## Improved synthesis of 3,3‴″-bis[4-(*tert*-butyldimethylsilyl)phenyl]sexiindene (4c) by Suzuki−Miyaura coupling

A solution of **8** (48 mg, 0.10 mmol), **7a** (110 mg, 0.23 mmol), $K_2CO_3$ (70 mg, 0.51 mmol), and Pd(PPh$_3$)$_4$ (5.9 mg, 5.1 μmol) in THF (2.0 mL) and $H_2O$ (0.4 mL) was stirred for 12 h at 75 °C in the dark. After cooling the reaction mixture to an ambient temperature, the resulting precipitate was collected by vacuum filtration and washed with $H_2O$ and ethyl acetate to give 48 mg of **4c** as dark green solids (45 μmol, 45% yield).

## Single-crystal X-ray diffraction

Single-crystal X-ray diffraction data were collected on a Rigaku XtaLAB AFC10 diffractometer equipped with FR-X generator, Varimax optics, and PILATUS 200 K photon counting detector with MoKα radiation (λ = 0.71073 Å) for **3a**, or on synchrotron radiation (λ = 0.4125−0.4137 Å) at the BL02B1 beamline in SPring-8 (JASRI) for **3b**, **4a**, **4b**, **6a**, **7a**, and **8**. The structure was solved by direct methods

(SHELXT 2018/2)[46] and refined by the full-matrix least-squares on $F^2$ (SHELXL-2018/3)[47]. All hydrogen atoms were placed using AFIX instructions, while all non-hydrogen atoms were refined anisotropically. Detailed measurement conditions and crystal data for these compounds are described in the Supplementary Methods in the Supplementary Information.

## Cyclic voltammetry measurements

Cyclic voltammetry (CV) measurements were performed with an ALS/chi-610E electrochemical analyzer (BAS). The CV cell consisted of a glassy carbon electrode, a Pt wire counter electrode, and an Ag/AgNO$_3$ reference electrode. The measurements were carried out under a N$_2$ atmosphere using a sample solution in THF (reduction) or $CH_2Cl_2$ (oxidation) with a concentration of 1 mM containing 0.1 M tetrabutylammonium hexafluorophosphate as a supporting electrolyte. The redox potentials were calibrated with a ferrocene/ferrocenium ion couple (Fc/Fc$^+$). The results are shown in Fig. 4a, Supplementary Figs. 5 and 6, and their data are summarized in Supplementary Table 7.

## Steady-state electronic absorption and photoluminescence measurements

UV/Vis/NIR absorption spectra were measured with a Shimadzu UV-3600 spectrophotometer with a resolution of 0.2 nm using dilute sample solutions in spectral grade $CH_2Cl_2$ in a 1 cm-square quartz cuvette. The results are summarized in Fig. 5a, Supplementary Fig. 7, and Supplementary Table 8. Photoluminescence spectra of **3a**−**c** and **4a**−**c** were attempted to be measured with a Hamamatsu Photonics Quantaurus-QY Plus calibrated integrating sphere system C13534-02 equipped with a high-power Xe lamp L13685-01, a near-infrared multichannel detector C13684-01, and a quartz-made Dewar vessel, using dilute sample solutions in 2-methyl tetrahydrofuran. The solvent was purified by vacuum distillation over CaH$_2$, and the resulting sample solutions were degassed by purging the argon gas stream for ca. 10 min. However, **3a**−**c** and **4a**−**c** were virtually non-luminescent in a degassed 2-methyltetrahydrofuran solution at an ambient temperature and 77 K.

## Quantum chemical calculations

The geometry optimizations for the oligo(biindenylidene)s **3a**−**c** and **4a′**−**c′**, the radical anions [**4a′**]$^{•−}$ and [**4b′**]$^{•−}$, dianions [**4a′**]$^{2−}$ and [**4b′**]$^{2−}$, and fullerene C$_{60}$ were carried out using the Gaussian16 Revision B.01[48] with default thresholds and algorithms. The oligo(biindenylidene) **4a′**, **4b′**, and **4c′** are the model compounds of **4a**, **4b**, and **4c** wherein *t*-BuMe$_2$Si groups were simplified to Me$_3$Si groups to reduce the computational costs. Initially, the geometry optimizations for **4a′**−**c′** were performed by using various types of density functionals, including B3LYP, CAM-B3LYP, PBE0, M06-2X, and ωB97XD functionals with the 6-31+G(d) basis set, and the optimized geometries and the orbital energy levels of Kohn−Sham HOMOs and LUMOs were compared (Supplementary Fig. 8). Based on these comparisons, we concluded that the PBE0 is the most suitable density functional among the ones examined, and therefore all the other calculations of **3a**−**c**, [**4a′**]$^{•−}$, [**4b′**]$^{•−}$, [**4a′**]$^{2−}$, [**4b′**]$^{2−}$, and C$_{60}$ were carried out at the (U)PBE0/6-31+G(d) level of theory. For the geometry optimizations of radical anions ([**4a′**]$^{•−}$ and [**4b′**]$^{•−}$) and dianions ([**4a′**]$^{2−}$ and [**4b′**]$^{2−}$), the optimized geometries for the corresponding charge-neutral species **4a′** and **4b′** were used as initial guesses. The stationary points were optimized without any symmetry assumptions and characterized by frequency analysis at the same level of theory (the number of imaginary frequencies was 0). The cartesian coordinates for the optimized geometries are given in Supplementary Tables 13–35. The selected C−C bond lengths of the optimized geometries of charge-neutral biindenylidene monomers **3a** and **4a′**, dimers **3b** and **4b′**, and trimers **3c** and **4c′** were summarized in Supplementary Tables 9−11 and Supplementary Fig. 10. Those for [**4b′**]$^{•−}$ and [**4b′**]$^{2−}$ were also

given in Fig. 4b and Supplementary Fig. 11. Mulliken spin densities of [4a′]·− and [4b′]·− were shown in Supplementary Fig. 12. The electrostatic potential map for the optimized geometry of [4b′]²⁻ was calculated at the MP2/6-31+G(d) level of theory and the result was shown in Fig. 4c. Time-dependent (TD)-DFT vertical excitation energy calculations for the optimized geometries of 3a−c and 4a′−c′ were performed at the same level of theory. The results are summarized in Fig. 5b, Supplementary Figs. 9 and 14, and Supplementary Table 12. The nucleus-independent chemical shift (NICS)[49–51] values of 4b and [4b′]²⁻ were calculated for the centroid of each ring using the GIAO method at the PBE0/6-31 + G(d) level of theory, and the results were summarized in Supplementary Fig. 13.

### Time-resolved microwave conductivity (TRMC) measurements[43,44]

3a, 3b, 3c, and 4c were deposited onto quartz plates in their polycrystalline states. The polycrystalline thin films were overcoated by Cytop®, and evacuated under vacuo for 1 h at 40 °C prior to TRMC measurements. The films were fixed into a $N_2$-filled microwave cavity with Q of ca. 2400 at -9 GHz, and excited at 355 nm of third harmonic generation from a Spectra-Physics INDI Nd:YAG laser. Transmittance of all the films was measured with a Ophir VEGA power meter with a PE-25 head. Transient conductivity signals were evolved through a Schottky diode, amplified, and monitored by a Tektronics TDS3054 digital oscilloscope. All the measurements were carried out at an ambient temperature.

### Transient absorption spectroscopy measurements

An identical polycrystalline film of 3c was used for the spectroscopy measurements. White light continuum from a 150 W Xenon lamp (Ushio) was used as an probe, and transmitted light through the film has been spectrally dispersed by a monochromater with focal plane of 300 or 150 mm (C11119-04, Hamamatsu Photonics). Dispersed spectral lines were guided into a high dynamic range streak camera (C13410-01A, Hamamatsu Photonics), amplified by an image intensifier (V12303, Hamamatsu Photonics), and detected with a digital CMOS camera (ORCA-Flash4.0). The 3rd harmonic generation at 355 nm from a nanosecond Nd:YAG laser (NT341A, EKSPLA) was used as an excitation light source. All the measurements were carried out at an ambient temperature.

### Electrochemical spectroscopy measurements

Spectroelectrochemical measurements were carried out with a custom-made optically transparent thin-layer electrochemical (OTTLE) cell (light pass length = 1 mm) equipped with a platinum mesh, a platinum coil, and a silver wire as the working electrode, the counter electrode, and the pseudo-reference electrode, respectively. Spectroscopy was carried out with a JASCO V-570 spectrophotometer, and the potential was applied with an ALS/chi Electrochemical Analyzer model 612A. All the measurements were carried out at an ambient temperature.

### Reporting summary

Further information on research design is available in the Nature Portfolio Reporting Summary linked to this article.

## Data availability

The crystallographic data (CIF files) for the structures reported in this Article have been deposited with the Cambridge Crystallographic Data Centre (CCDC), under deposition numbers CCDC 2184333 (3a), 2184328 (3b), 2184330 (4a), 2184329 (4b), 2184334 (6a), 2184332 (7a), and 2184331 (8). These data can be obtained free of charge from CCDC at [www.ccdc.cam.ac.uk/data_request/cif]. All experimental data within the article and its Supplementary Information are available from the corresponding authors upon request. Source data are available as a Source data file. Source data are provided with this paper.

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

## Acknowledgements

The authors thank the iCeMS Analysis Center for providing access to the NMR spectrometer. The authors are grateful to Prof. Susumu Kitagawa (Kyoto University) for providing access to the X-ray diffractometer. Mass spectrometry experiments were carried out at the MS Section in the Institute for Chemical Research, Kyoto University. Synchrotron X-ray crystallography experiments were performed at the BL02B1 beamline of SPring-8, with the approval of the Japan Synchrotron Radiation Research Institute (JASRI; Project No. 2019A1057, 2019B1129, 2019B1784, 2020A0557, 2020A1056, 2020A1644, 2020A1650, 2020A1656, 2021A1578, 2021A1592, 2021B1435, 2021B1798, 2021B1833, and 2022A1705). The authors also thank Ms. Mika Sakai (Nagoya University) for her assistance in the measurement of photophysical properties. This research was funded by a Grant-in-Aid for Scientific Research (B) (JSPS KAKENHI Grant Number 21H01916 for A.F.), Grant-in-Aid for Transfor-mative Research Areas (A) "Condensed Conjugation" (JSPS KAKENHI Grant Numbers 20H05864 for A.F. and 20H05862 for S.S.), and Grant-in-Aid for Scientific Research (A) (JSPS KAKENHI Grant Number 18H03918 for S.S.) from MEXT, Japan, and research promotion grant from the Kyoto University Foundation (A.F.). M.H. thanks JSPS for the Research Fellowship for Young Scientists.

## Author contributions

A.F. conceived the concept. A.T. explored the initial synthetic route. M.H. and N.S. performed the synthesis and characterization. S.T. and Y.M. assisted with synthesis and characterization. M.H. conducted crystallographic analysis and quantum chemical calculations. S.S. evaluated electron conductivity. M.H. and A.F. prepared the manuscript. S.Y. proofread it. A.F. directed the project.

## Competing interests

The authors declare no competing interests.
