## [Peer Review File · Nature Communications]

One-dimensional fragments of fullerene C60 that exhibit robustness toward multi-electron reductionReviewers' Comments:

Reviewer #1:

Remarks to the Author:

Fukazawa and coworkers synthesized a series of oligo(biindenylidene)s, which are noncurved one-dimensional fragment of fullerene C₆₀, and Suzuki-Miyaura coupling was significantly superior to Negishi coupling in the synthetic approach. The structures of most compounds (except 3c and 4c) were verified by X-ray crystallography and DFT calculations. The most interesting part of this study should be the electrochemical properties that studied compounds are multi-electron acceptors and the number of electrons accepted is equal to the number of five-membered rings in their main chains. The studied compounds exhibited ultraviolet/visible/near-infrared absorption extending to around 800 nm. This manuscript could be recommended to publish in Nature Communications. Two main points: 1) The bond lengths of the polyene system in all members should be compared, namely 4a' vs 4b' vs 4c', and their anions. 2) The driving forces for the reduction process should be explained.

Reviewer #2:

Remarks to the Author:

In the manuscript titled "One-dimensional fragments of fullerene C₆₀ that exhibit robustness toward multi-electron reduction and pronounced light absorption" by M. Hayakawa et al. a series of oligo biindenylidenes are synthesized and characterized. It is found that the compounds can undergo multi-electron reduction, which is correlated to the size of the oligomer, and that they possess good optical absorption extending through the visible wavelength range. These compounds are contrasted against C₆₀, which is a common fullerene found in many inorganic and organic electronic device applications. Overall, the findings regarding the oligo biindenylidenes are interesting, the manuscript contains sufficient detail allowing for reproducibility, and the conclusions are straightforward. However, the synthetic methods used are well established and although the multi-electron reduction of the oligo biindenylidenes is interesting the utility of these compounds for inorganic/organic electronic applications was not demonstrated, which significantly limits the impact of this study. Additionally, there are areas that could benefit from further elaboration and/or experiments, which are detailed in the comments below. Since the impact and utility for oligo biindenylidenes has not been adequately demonstrated, this manuscript is not recommended for publication in Nature Communications.

Comments:

1. The use of the indene dimer (5) is very creative, since the synthesis was reported decades ago (ref. 38) but it was never widely studied. However, the yields for the Negishi cross coupling (3a-3c, 4a-4c, 6a/b, and 7a/b) are very low. Additionally, the yields for 6a and 7a, which are needed for 3c/4c, are < 20% limiting the practicality of these compounds in comparison to fullerenes, which are commercially available and are relatively low-cost electron transport materials. Did the authors attempt other cross-coupling methodologies, e.g. Suzuki-Miyaura, for the synthesis of 6a/7a? What parameters were optimized to improve the yield? The authors should provide this info in SI and briefly comment on it in the manuscript to show the chemistry was adequately investigated.

2. In Fig. 3, the authors show the single-crystal structures for 3a and 4b. These findings show curvature along the conjugated backbone, indicating that these compounds are not planar or flattened as described throughout the manuscript. These structural motifs may play an important part in the solid-state electron transport properties, and so they should not be overlooked, since many small-molecule non-fullerene electron transport materials adopt similar conformations. It is recommended the authors rephrase the statements regarding planarity or flattening to be more accurate/transparent. Additionally, distances between oligomers should be provided to see if there is evidence of pi-pi interactions, and the authors should comment on the influence of the end-caps, phenyl or TMS-phenyl, in the molecular packing.

3. The electrochemistry and the optical absorption for the biindenylidene oligomers is fascinating. However, the manuscript would significantly benefit from the structure-property relationships being succinctly summarized as was done for fullerene in Fig. 1a. More importantly, the cyclic-voltammetry

measurements for 3a-c should also be provided. If the compounds are solubility limited, then authors should test the solubility limits in a wide variety of solvents and comment on their findings in the manuscript. The authors should also provide the energy levels (HOMO/LUMO) versus vacuum, which can be easily calculated using the estimates provided from CV and compare these to the calculated values in Fig. 5b. For the optical absorption measurements, the authors should include the optical bandgaps in Table S2. It is mentioned that 3a-c and 4a-c did not display any fluorescence. Was phosphorescence detected? The authors should include these measurements and comment on their findings.

4. A central feature of fullerenes are the isotropic electron transport properties, and for C60 it can also be easily vacuum deposited to form interfacial layers in various inorganic/organic electronic devices. Although these biindenylidene oligomers show multi-electron reduction the electron transport properties need to be demonstrated. The authors should provide the electron mobility (SCLC and/or OFET) for 3a-c/4a-c to demonstrate the impact and utility. From the crystal structures, it can be seen these compounds have a preferred orientation, and so charge-transport may be anisotropic and occur preferentially in the vertical or horizontal directions, which the authors should also comment on.

Reviewer #3:
Remarks to the Author:
review attached

This work is novel; it is important, and it will have an impact significantly beyond just the field of organic chemistry. The authors are to be congratulated for having created a new family of organic electron acceptors.

For organic solar cells, organic light-emitting diodes, and other devices that require both p-type and n-type charge-carrying materials, scientists are constantly searching for molecules or materials that can serve as better electron donors or better electron acceptors. This paper introduces a brand-new class of electron acceptors.

The most common strategy for designing electron accepting molecules is to decorate known molecules with strongly electron-withdrawing functional groups, e.g., perylene diimides. More electron-withdrawing groups generally lower the energy of the lowest unoccupied molecular orbital (LUMO) and thereby yield better electron acceptors. C_{60} and other fullerenes are unique among electron acceptors by virtue of having no electron-withdrawing functional groups. The authors of this paper have conceptually extracted from the C_{60} structure a string of 5-membered rings (Fig. 1b), synthesized the first three of resulting biindenylidene oligomers (Fig. 2), and shown experimentally that they are exceptionally good electron acceptors (Fig. 4a). The first member of the family has two 5-membered rings and reversibly accepts two electrons; the second member of the series has four 5-membered rings and reversibly accepts four electrons; the third oligomer in the series has six 5-membered rings and reversibly accepts six electrons. There is every reason to believe that longer oligomers in this series will accept as many electrons as there are 5-membered rings in the structure. Few organic molecules other than C_{60} and the larger fullerenes are capable of accepting as many as six electrons reversibly.

One of the most remarkable features of these new electron acceptors is that they are hydrocarbons, i.e., they are not molecules decorated with strongly electron-withdrawing functional groups. Thus, the authors have diverged from the customary strategy for designing electron accepting molecules and have introduced a new paradigm.

Unlike C_{60} , these new biindenylidene oligomers are very good at absorbing visible light (Fig. 5a). Molar extinction coefficients up to 60,000 are reported in their visible absorption spectra. By contrast, C_{60} absorbs visible light only very weakly, with a molar extinction coefficient of less than 1000. The long wavelength HOMO-LUMO transition in C_{60} is symmetry forbidden, but that is not the case for these new biindenylidene oligomers. This enormous difference in photophysical properties between C_{60} and these new biindenylidene oligomers presents opportunities for designing optoelectronic devices that have been impractical using C_{60} .

The HOMO-LUMO gap of C_{60} is only 2.16 eV, but the new biindenylidene oligomer with four 5-membered rings has an even smaller HOMO-LUMO gap (1.96 eV), and for the oligomer with six 5-membered rings it is smaller still (1.72 eV) (Fig. 5b). The popularity of C_{60} and its derivatives as electron acceptors (in organic solar cells, for example) stems from its exceptionally low-lying LUMO (3.63 eV); however, the new biindenylidene oligomer with six 5-membered rings has a LUMO lying almost as low (3.41 eV) (Fig. 5b), and longer oligomers may match or even surpass C_{60} .

Multiple X-ray crystal structures provide atomic-scale insight into the structures and shapes of these new biindenylidene oligomers and several of the intermediates used in their synthesis. The supplementary information is thorough and excellent.

Minor issues:

- 1) Figure 5 caption: "Inset: magnified spectra..." should be singular "Inset: magnified spectrum..."
- 2) Reference 26: The first author's name is Bronstein (not Cronstein).

Reviewer #4:

Remarks to the Author:

This manuscript reports the synthesis of oligo(biindenylidene)s with the goal to understand the origin on the redox capacities of C₆₀. This has required significant synthetic efforts. The rationale presented by the author is supported by the experimental evidence. The paper goes to the point, is well written. However, I feel it is quite limited in scope and I feel it is more suited for a Chemistry journal such as *Angew. Chem* or *JACS* rather than here. I have only two minor points. 1. The authors stress the results are obtained in the absence of "curvature". But the X-ray of 4b has a curvature, which is not insignificant. 2. Do the computations reproduce the structure of at least 3a and b? By looking at the image of fig 4 it seems not. But, this should be clarified. Do the computed structure vary significantly going from the neutral to the redox species?

We appreciate the reviewers for their careful reviews and constructive comments on our manuscript. Our point-to-point response to the reviewers' comments and what has been revised are listed as follows.

Responses and changes made to Reviewer #1

(changes in the manuscript are highlighted in yellow)

The authors appreciate Reviewer 1 for the positive comments and several constructive suggestions, which we have addressed as described below thoroughly.

[Comment #1]

The bond lengths of the polyene system in all members should be compared, namely 4a' vs 4b' vs 4c', and their anions.

Reply & Revisions: The comparison of the bond lengths of the polyene substructures of charge-neutral species **3a**, **3b**, **4a**, and **4b** obtained by crystallographic analysis has already been shown in Supplementary Table 1 in the original version. The bond lengths of trimers **3c** and **4c** could not be compared experimentally since all the attempts to obtain single crystals suitable for X-ray crystallographic analysis remain unsuccessful even after intensive investigations. To enable comparison of the C–C bond lengths including trimers, those in the optimized geometries by DFT calculations have been added to Supplementary Tables 9–11. In addition, the comparison of the bond lengths was added to Supplementary Fig. 10. These comparisons provide the following insights into the structures of charge-neutral oligo(biindenylidene)s.

- 1) Comparison of the bond lengths obtained by DFT calculations at the PBE0/6-31+G(d) level of theory for the monomers (**3a** and **4a'**) and dimers (**3b** and **4b'**) with those determined by crystallographic analysis shows that the errors are all within 0.01 Å, indicating that the present computational methods sufficiently reproduced the experimental structure.
- 2) The C–C bond lengths in **3a–c** and **4a'–c'** along the polyenyl substructure do not have any noticeable dependence on the chain lengths, as already been described in the section of "Synthesis and characterization" in the manuscript based on the results of X-ray crystallographic analyses.

Supplementary Table 9: Selected Bond Lengths [Å] in the Optimized Geometries of Charge-Neutral Biindenylidene Monomers **3a** and **4a'** together with the Corresponding Experimental Data

3a (R = H)
4a (R = SiMe₂t-Bu)
4a' (R = SiMe₃)

Bond	3a				4a'		4a			
	Calc. ^[a]	Exp. ^[b,c]		Calc. ^[a]	Exp. ^[b]					
		unit A	unit B							
a	C11–C12	1.36434	C2–C3	1.355(2)	C32–C33	1.356(2)	C15–C18	1.36636	C2–C3	1.362(2)
	C25–C26	1.36565	C11–C12	1.353(2)	C41–C42	1.357(2)	C3–C21	1.36636	C11–C12	1.366(2)
b	C12–C14	1.45085	C1–C2	1.457(2)	C31–C32	1.456(2)	C14–C18	1.45004	C1–C2	1.450(2)

	C26–C28 1.45081	C10–C11 1.454(2)	C40–C41 1.452(2)	C6–C21 1.45004	C10–C11 1.455(2)
c	C14–C28 1.37465	C1–C10 1.363(2)	C31–C40 1.370(2)	C6–C14 1.37459	C1–C10 1.373(2)

[a] Computational results by geometry optimization calculated at the PBE0/6-31+G(d) level of theory. [b] Experimental values determined by X-ray crystallographic analyses. [c] Compound **3a** consists of two crystallographically independent units A and B in the crystal lattice.

Supplementary Table 10: Selected Bond Lengths [Å] in the Optimized Geometries of Charge-Neutral Biindenylidene Dimers **3b** and **4b'** together with the Corresponding Experimental Data

3b (R = H)
4b (R = SiMe₂t-Bu)
4b' (R = SiMe₃)

Bond	3b		4b'		4b	
		Calc. ^[a]	Exp. ^[b]		Calc. ^[a]	Exp. ^[b]
a	C11–C12	1.36696	1.361(3)	C11–C12	1.36746	1.364(7)
	C53–C54	1.36696		C44–C46	1.36746	
b	C12–C14	1.44850	1.455(2)	C12–C14	1.44814	1.448(6)
	C54–C56	1.44850		C43–C44	1.44814	
c	C14–C28	1.37687	1.381(3)	C14–C15	1.37703	1.381(7)
	C42–C56	1.37687		C42–C43	1.37703	
d	C26–C28	1.44518	1.446(2)	C15–C16	1.44488	1.452(7)
	C40–C42	1.44518		C40–C42	1.44488	
e	C25–C26	1.37292	1.375(3)	C16–C18	1.37320	1.364(7)
	C39–C40	1.37292		C39–C40	1.37320	
f	C25–C39	1.44211	1.449(3)	C18–C39	1.44185	1.457(10)

[a] Computational results by geometry optimization calculated at the PBE0/6-31+G(d) level of theory. [b] Experimental values determined by X-ray crystallographic analyses.

Supplementary Table 1. Selected Bond Lengths [Å] in the Optimized Geometries of Charge-Neutral Biindenylidene Trimers **3c** and **4c'**^[a]

3b (R = H)
4b' (R = SiMe₃)

Bond	3c	4c'
a	C39–C40 1.36554	C44–C46 1.36768
	C81–C82 1.36546	C67–C68 1.36743
b	C40–C42 1.44850	C43–C44 1.44770
	C82–C84 1.44875	C68–C70 1.44802
c	C42–C56 1.37751	C42–C43 1.37741
	C70–C84 1.37725	C70–C71 1.37710
d	C54–C56 1.44484	C40–C42 1.44422
	C68–C70 1.44534	C71–C72 1.44492
e	C53–C54 1.37331	C39–C40 1.37392
	C67–C68 1.37285	C72–C74 1.37325

f	C11–C53 1.44096	C18–C39 1.44067
	C25–C67 1.44260	C11–C74 1.44250
g	C11–C12 1.37408	C16–C18 1.37440
	C25–C26 1.37277	C11–C12 1.37296
h	C12–C14 1.44297	C15–C16 1.44266
	C26–C28 1.44272	C12–C14 1.44247
i	C14–C28 1.38021	C14–C15 1.38033

[a] Computational results by geometry optimization calculated at the PBE0/6-31+G(d) level of theory.

Supplementary Fig. 10: Plot of selected bond lengths in the optimized geometries of charge-neutral oligo(biindenylidene)s **3a–c** (a) and **4a'–4c'** (b) calculated at the PBE0/6-31+G(d) level of theory. The bond numbers *a–i* correspond to those illustrated in Supplementary Tables 3–5.

As for the geometries of anionic species, the results of geometry optimizations of the radical anions (**3^{•-}** and **4^{•-}**) and dianions (**3²⁻** and **4²⁻**) for a series of oligo(biindenylidene)s, and the spin density distribution of the radical anions (**4a^{•-}** and **4b^{•-}**) have already been given in Fig. 4, and Supplementary Figs. 11, and 12 in SI (Supplementary Figs. 7 and 8 in the original version). However, we refrained from the discussion about radical anions (**3b^{•-}**, **4b^{•-}**, **3c^{•-}**, and **4c^{•-}**) in this manuscript, since there were discrepancies between the character of the optimized geometries/spin density distributions and the experimental results of electrochemistry. Given the experimental results of electrochemical and photophysical studies, we currently hypothesize that these discrepancies are most likely due to the existence of several conformers in the solution. While further investigations to overcome this discrepancy are needed, we feel that such investigations including computational studies of all possible conformers into consideration, and experimental studies for the structural characterization of reduced species, are beyond the scope of the current manuscript.

To clarify the above-mentioned situation, the discussion regarding the discrepancy between computation and experiments for radical anions has been described in part in the original version of SI as Supplementary Discussion 3 (Supplementary Discussion 6 in the revised version), which was further revised as follows. The newly added texts are underlined.

Supplementary Discussion 6: Insights into the Structural Changes of Oligo(biindenylidene)s Upon Reduction. The selected C–C bond lengths of the optimized geometries of charge-neutral biindenylidene monomers and dimers **4a'** and **4b'**, and the corresponding radical anions $[4a']^-$ and $[4b']^-$, and dianions $[4a']^{2-}$ and $[4b']^{2-}$ were summarized in Supplementary Fig. 11. Those for **4b** and $[4b]^{2-}$ were also shown in Fig. 4.

The optimized structures of the radical anions $[4a']^-$ and $[4b']^-$ adopted highly coplanar geometries with significantly smaller bond length alternation (BLA) along the main chain (Supplementary Fig. 11), indicative of the delocalization of spin and charge densities along the main chain. The spatial distribution of the Mulliken spin densities of $[4a']^-$ and $[4b']^-$ supports the delocalization of spin densities (Supplementary Fig. 8). Given that these optimized structures of these radical anions $[4a']^-$ and $[4b']^-$ differ significantly from those of the charge-neutral species **4a'** and **4b'**, the potential difference between the one- and two-electron reduction of oligo(biindenylidene)s should be reasonably large. Indeed, the energy split between the first and second redox waves of the monomer **4a** was 0.31 V in cyclic voltammetry (Fig. 4a).

On the other hand, for dimer **4b**, the splitting between the first and second redox waves is as small as 0.15 V (Fig. 4a), which is inconsistent with the above working hypothesis based on the optimized geometry of $[4b']^-$. This discrepancy may be attributable to the presence of a conformer of **4b** in solution with varied dihedral angles around the central C–C bond (bond *f*). The broad first absorption band observed in the electronic absorption spectrum (Fig. 5a and Supplementary Fig. 7) strongly suggests that **4b** is a mixture of various conformers in solution at an ambient temperature. The reduction of the mixture of several conformers of **4b** might produce several kinetically favorable conformers of $[4b']^-$ with distinct structures from the optimized geometry. Similarly, we encountered similar issues in the DFT calculations for radical anionic trimer $[4c']^-$ and dianionic trimer $[4c']^{2-}$ most likely due to many possible conformers. Given the difficulty in obtaining computational results that provide a reasonable explanation for experimental results at this stage, we have refrained from the discussion about radical anions of oligo(biindenylidene)s (**3^{•-}** and **4⁻**) and the reduced species of trimers (**3c** and **4c**) in this manuscript based on the computational results. More in-depth investigations by both experiments and theoretical calculations will be reported elsewhere.

In addition, since this Supplementary Discussion was not cited in the manuscript, the citation was added at the end of the section on “Electrochemical properties” on page 9 as follows:

This charge separation in the dianion may be responsible for decreasing the peak separation of the first and second reduction waves as the chain length increases and reflects the structural feature of one-dimensional π -conjugated chains (see also Supplementary Discussion 6 for more detailed discussions including the results of radical anions $[4b']^{2-}$).

[Comment #2]

The driving forces for the reduction process should be explained.

Reply & Revisions: At this stage, we assume that the driving force for the multistep reduction is acquiring the aromaticity of five-membered ring moieties in addition to moderately low-lying π^* orbitals caused by cross-conjugation. To clarify this perspective, the manuscript was amended as follows.

(In the second paragraph on page 2)

Several structural factors have been proposed, including the degeneracy of LUMO and LUMO+1 due to the highly symmetrical structure, the mitigation of bond angle strain around the carbon atoms upon

reduction due to the inherently pyramidalized geometry, and the presence of five-membered ring substructures that can acquire Hückel aromaticity in a reduced state (Fig. 1a).^{8,20-21}

(In the first paragraph on page 9)

Moreover, the bond-length equalization as shown in Fig. 4b and Supplementary Fig. 9b as well as the negative values of the nuclear-independent chemical shifts (NICS, Supplementary Fig. 13) for the terminal indenylidene moieties in [4b]²⁻ strongly indicate the 10 π Hückel aromaticity of the two indenyl anion moieties. Given these computational results, the localization of electron densities at both termini should be ascribed to the minimization of the electronic repulsion and the stabilization caused by the aromaticity of the two indenyl anion moieties.

Responses and changes made to Reviewer #2

(changes in the manuscript are highlighted in green)

We thank Reviewer 2 for the constructive criticism. While the findings in this study may serve as a basis for designing new non-fullerene electron acceptors in the near future, the main focus of this paper is not on the creation of alternative materials to fullerene in electronic device applications. Nevertheless, we are convinced that this paper is of sufficient fundamental scientific significance and provides findings that should overturn conventional perceptions.

[Comment #1]

The use of the indene dimer (5) is very creative, since the synthesis was reported decades ago (ref. 38) but it was never widely studied. However, the yields for the Negishi cross coupling (3a-3c, 4a-4c, 6a/b, and 7a/b) are very low. Additionally, the yields for 6a and 7a, which are needed for 3c/4c, are < 20% limiting the practicality of these compounds in comparison to fullerenes, which are commercially available and are relatively low-cost electron transport materials. Did the authors attempt other cross-coupling methodologies, e.g. Suzuki-Miyaura, for the synthesis of 6a/7a? What parameters were optimized to improve the yield? The authors should provide this info in SI and briefly comment on it in the manuscript to show the chemistry was adequately investigated.

Reply & Revisions: Although we initially reported the improved synthesis based on Suzuki-Miyaura coupling only for trimers **3c** and **4c**, all synthetic routes for the other oligomers **3a-b** and **4a-b** were also reexamined by Suzuki-Miyaura coupling based on critical comments about the low yield of Negishi coupling. Accordingly, the results of only Suzuki-Miyaura coupling are now shown in Fig. 2, and the previous results for the Negishi coupling have been moved to SI. Together with them, we also added further discussions based on the results of our synthetic investigations of target compounds based on several transition metal-catalyzed cross-coupling reactions in SI as Supplementary Discussion 1. Now the paragraph in the manuscript reads:

(In the section of "Synthesis and characterization" on page 4)

Oligo(1,1'-biindenylidene)s **3** and **4** were synthesized by iterative cross-coupling reactions using 3,3'-dibromo-1,1'-biindenylidene (**5**)^{38,39} as a precursor. The Pd-catalyzed coupling reaction of **5** with phenylzinc chloride under typical Negishi-coupling conditions resulted in the formation of a complex mixture containing both the expected cross-coupling products, such as monophenylated biindenylidene **6a** and diphenylated biindenylidene **3a**, and the unexpected oligomers such as the monophenylated biindenylidene dimer **6b**, diphenylated biindenylidene dimer **3b**, and trimer **3c** (Supplementary Discussion 1 and Supplementary Fig. 2). Conversely, the Suzuki-Miyaura coupling using organoboronates significantly improved the selectivity of the cross-coupling reaction over the undesirable homo-coupling processes (Fig. 2). Specifically, the reactions of **5** with 2.4 equivalents of phenylboronic acid or 4-(*tert*-butyldimethylsilyl)phenylboronate in the presence of K₂CO₃ and a catalytic amount of Pd(PPh₃)₄ successfully produced the end-capped biindenylidenes **3a** and **4a** in 62% and 80% yields, respectively (Fig. 2, top). Similarly, the Suzuki-Miyaura coupling of **5** with the reduced amount of aryl boronates afforded monoarylated bromobiindenylidenes **6a** and **7a**, which are useful precursors for the iterative synthesis of biindenylidene oligomers, in moderate yields without the formation of homo-coupling products (Fig. 2, middle). The bromobiindenylidenes **6a** and **7a** were

subjected to consecutive Miyaura-Ishiyama borylation and Suzuki-Miyaura coupling processes in one-pot in the presence of bis(pinacolato)diboron, Pd(PPh₃)₄, and potassium carbonate to afford the corresponding biindenylidene dimers **3b** (70% yield) and **4b** (55% yield), respectively (Fig. 2, middle). The trimers **3c** and **4c** were synthesized by the transformation of dibromobiindenylidene **5** to the corresponding diboronate **8** by the Miyaura-Ishiyama borylation (see also Supplementary Discussion 2), followed by the Suzuki-Miyaura coupling with the excess amount of monobromide **6a** or **7a**, respectively (Fig. 2, bottom).

Fig. 2. Synthesis of biindenylidene oligomers.

The information given in the Supporting Information now reads:

Supplementary Discussion 1: Attempts for Cross-Coupling Reactions of 5 under the Kumada-Tamao-Corriu and Negishi Conditions. We initially examined the reactivity of **5** under the conditions of Kumada-Tamao-Corriu coupling with phenylmagnesium bromide by using Ni or Pd catalysts. However, all the reaction conditions resulted in the formation of a black-colored complex mixture without the formation of desired phenylated **3a** and **6a**. Given that a similar result was obtained even with the addition of phenylmagnesium bromide to **5** in the absence of any catalyst precursors, we assume that the single-electron transfer from Grignard reagent to **5** followed by the decomposition of the resulting radical anion species took place due to the highly electron-accepting character of **5**.

Assuming that undesirable electron transfer reaction from arylmetal reagents to **5** should be suppressed for successful cross-coupling reactions, we next investigated the conditions for a typical Negishi cross-coupling reaction. Initially, we attempted the synthesis of monoarylated 1,1'-biindenylidenes (**6a** and **7a**) as end-capping building blocks by Pd-catalyzed Negishi coupling of dibromide **5** with arylzinc chloride (Supplementary Fig. 2). However, careful separation of the products provided the expected cross-coupling products, such as monophenylated biindenylidene **6a**

(15% yield) and diphenylated biindenylidene **3a** (4% yield), and the unexpected oligomers including the monophenylated biindenylidene dimer (**6b**, 1% yield), diphenylated biindenylidene dimer (**3b**, 1% yield), and trimer (**3c**, 0.9% yield). The reaction using trialkylsilyl-substituted phenylzinc chloride gave results identical to those described above, and the corresponding oligo(biindenylidene)s with silyl groups at both termini (**4a–c**) were obtained (Supplementary Fig. 2).

Supplementary Fig. 2: Reaction of **5** with arylzinc reagents.

We also added the details about the Miyaura-Ishiyama borylation of dibromide **5** to diboronate **8** as Supplementary Discussion 2 in the supporting information. The discussion now reads:

Supplementary Discussion 2: Miyaura-Ishiyama Borylation Reactions. For the synthesis of oligo(biindenylidenes) by iterative cross-coupling reactions, the biindenylidenes bearing organometallic functional groups such as boronic acid or boronate are essential precursors. However, preliminary investigations indicated that the conventional transformation reactions of aryl and/or vinyl halides by halogen-metal exchange reaction followed by the trapping with electrophiles are not feasible for dibrominated **5** because of the low solubility and the concomitant electron transfer reactions (see also Supplementary Discussion 1). Therefore, among the already known transformations of C(sp²)-Br bond to the corresponding organoboronates, we focused on Miyaura-Ishiyama borylation,^{S1} which does not require strong reducing reagents or low-temperature conditions. In the presence of bis(pinacolato)diboron (B₂pin₂) as a boron source, several different reaction conditions including a Pd source, ligand, base, and solvent were examined (Supplementary Table 1). The use of PdCl₂(dppf), the most commonly used catalyst precursor for Miyaura-Ishiyama borylation, did not give the desired product **8** regardless of the choice of base, solvent, or temperature, albeit complete consumption of **5** (entries 1–6). After the exhaustive screenings of Pd source and ligands, only Pd(PPh₃)₄ was found to give the desired product **8** (entries 18 and 19) in contrast to several other catalyst precursors such as Pd(OAc)₂ (entries 7–11), Pd₂(dba)₃ (entries 12 and 13), PdCl₂(PPh₃)₂ (entry 15), and XPhos Pd G3 (entries 16 and 17). Specifically, **8** was obtained in 40% yield when potassium acetate and THF were used as a base and solvent (entries 18 and 19), whereas no target product was observed with potassium phenoxide (entry 21). In addition, while the higher reaction temperature improved the yield of **8** in THF (entries 18 and 19), **8** was not obtained under heating in 1,4-dioxane at 110 °C (entry 20). Control experiments in the absence of B₂pin₂ showed that diboronate **8** is gradually decomposed under reaction conditions. The lack of stability of **8** might be partly responsible for the low yielding of **8** in only 40%. Given these results, we selected the *in-situ* preparation of the monoboronates derived from **6a** and **7a** and successive Suzuki-Miyaura coupling

for the synthesis of biindenylidene dimers **3b** and **4b**. This strategy was successful as can be seen in the reasonably high yields of **3b** (70%) and **4b** (55%).

- S1 Ishiyama, T., Murata, M. & Miyaura, N. Palladium(0)-Catalyzed Cross-Coupling Reaction of Alkoxydiboron with Haloarenes: A Direct Procedure for Arylboronic Esters. *J. Org. Chem.* **60**, 7508–7510 (1995).
- S2 Cardona, C. M., Li, W., Kaifer, A. E., Stockdale, D. & Bazan, G. C. Electrochemical considerations for determining absolute frontier orbital energy levels of conjugated polymers for solar cell applications. *Adv. Mater.* **23**, 2367–71 (2011).

Supplementary Table 1. Optimization of reaction conditions for Miyaura-Ishiyama borylation of **5**^a

Entry	Pd cat.	Ligand	Base	Solvent	Temp.	Yield
1	PdCl ₂ (dppf)	none	KOAc	DMSO	80 °C	n.d.
2	PdCl ₂ (dppf)	none	KOAc	1,4-dioxane	80 °C	n.d.
3	PdCl ₂ (dppf)	none	KOAc	THF	80 °C	n.d.
4	PdCl ₂ (dppf)	none	KOAc	toluene	110 °C	n.d.
5	PdCl ₂ (dppf)	none	KOPh	DMSO	50 °C	n.d.
6	PdCl ₂ (dppf)	None	KOPh	toluene	110 °C	n.d.
7	Pd(OAc) ₂	SIPr	KOAc	toluene	110 °C	n.d.
8	Pd(OAc) ₂	JohnPhos	Et ₃ N	1,4-dioxane	100 °C	n.d.
9	Pd(OAc) ₂	DPEPhos	Et ₃ N	1,4-dioxane	100 °C	n.d.
10	Pd(OAc) ₂	JohnPhos	KOPh	toluene	50 °C	n.d.
11	Pd(OAc) ₂	DPEPhos	KOPh	toluene	50 °C	n.d.
12	Pd(OAc) ₂	XantPhos	KOPh	toluene	50 °C	n.d.
13	Pd ₂ (dba) ₃	PCy ₃	KOAc	1,4-dioxane	80 °C	n.d.
14	Pd ₂ (dba) ₃	PCy ₃	KOPh	toluene	50 °C	n.d.
15	PdCl ₂ (PPh ₃) ₂	PPh ₃	KOPh	toluene	80 °C	n.d.
16	XPhos Pd G3	none	KOAc	toluene	110 °C	n.d.
17	XPhos Pd G3	none	KOPh	toluene	50 °C	n.d.
18 ^b	Pd(PPh ₃) ₄	none	KOAc	THF	50 °C	16%
19 ^b	Pd(PPh ₃) ₄	none	KOAc	THF	75 °C	40%
20 ^b	Pd(PPh ₃) ₄	none	KOAc	1,4-dioxane	110 °C	n.d.
21 ^b	Pd(PPh ₃) ₄	none	KOPh	THF	75 °C	n.d.

^aB₂pin₂ (3 equiv.), Pd catalyst (5 mol%), ligand (5 mol%), and base (2 equiv.) were used. ^bB₂pin₂ (2.4 equiv.), Pd catalyst (2 mol%), and base (5 equiv.) were used.

S1 Ishiyama, T., Murata, M. & Miyaura, N. Palladium(0)-Catalyzed Cross-Coupling Reaction of Alkoxydiboron with Haloarenes: A Direct Procedure for Arylboronic Esters. *J. Org. Chem.* **60**, 7508–7510 (1995).

In line with the above-mentioned revisions, Yu Matsuo was added as one of the co-authors because of his contribution to the synthetic studies.

The results of these synthetic investigations highlight the difficulty in the chemical transformation of redox-active pentafulvalene derivatives. Although each synthetic process may still contain room for further improvements, we believe that the revised manuscript now provides sufficient information that adequate synthetic investigations have been conducted.

[Comment #2-1]

In Fig. 3, the authors show the single-crystal structures for 3a and 4b. These findings show curvature along the conjugated backbone, indicating that these compounds are not planar or flattened as described throughout the manuscript. These structural motifs may play an important part in the solid-state electron transport properties, and so they should not be overlooked, since many small-molecule non-fullerene electron transport materials adopt similar conformations. It is recommended the authors rephrase the statements regarding planarity or flattening to be more accurate/transparent.

Revisions: Based on the reviewer's comment, we rephrased the statements "curvature" to "pyramidalization of carbon atoms" as follows to discriminate the geometry of each sp² carbon atom from the conformation of a molecular scaffold. The texts now read as follows:

(page 2, 2nd paragraph)

Hence, the minimum structural basis for the high electron affinity and exceptional stability toward the multi-electron reduction of fullerenes are of interest. Several structural factors have been proposed, including the degeneracy of LUMO and LUMO+1 due to the highly symmetrical structure, the mitigation of bond angle strain around the carbon atoms upon reduction due to the inherently pyramidalized geometry, and the presence of five-membered ring substructures (Fig. 1a).^{8,20–21} Although π -conjugated hydrocarbons with fragment structures of fullerenes can be promising for understanding the effect of each factor, most fullerene fragment molecules reported to date, such as corannulene,^{22,23} sumanene,²⁴ and larger molecules,^{25–30} have bowl-shaped structures in which most of the carbon atoms adopt pyramidalized geometries (Supplementary Fig. 1).

(page 3, 1st paragraph)

Therefore, to elucidate the role of the five-membered ring substructures in the exceptional electron affinity of fullerenes without the influence of high symmetry and pyramidalized geometries of carbon atoms, we conceived a molecular design of π -conjugated oligomers **3** and **4** composed of a one-dimensional fragment of C₆₀ in their main chains (Fig. 1c) based on the following idea. First, we focused on the hoop-shaped substructure of C₆₀ wherein 6 five-membered rings were linearly connected, that is, cyclic ter(1,1'-biindenylidene) **1** (Fig. 1b). Next, based on the similarity of the π -conjugated hoops and corresponding linear π -conjugated polymers, excluding their symmetry and structural distortion,^{32,33} we designed linear π -conjugated polymer **2** with an identical repeating unit to eliminate the pyramidalization effect.

(page 6, the last sentence)

These results demonstrate that oligo(biindenylidene)s are suitable for studying the effect of five-membered rings on the electron affinity of π -conjugated hydrocarbons without the influence of the pyramidalization of carbon atoms.

In addition, to better clarify the geometries of the carbon atoms in oligo(biindenylidene)s in a quantitative manner, the sum of bond angles and the σ - π interorbital angles ($\theta_{\sigma\pi}$) of the carbon atoms in five-membered rings were calculated for the crystal structures, and newly added in Supplementary Tables 3–6. Accordingly, the following descriptions were added in the discussion of the X-ray crystal structures.

(page 6, 9th line from the bottom)

More specifically, the π -orbital axis vector (POAV) analysis⁴¹ for the crystal structures of **3b** and **4b** revealed that the averaged values of the σ - π interorbital angle ($\theta_{\sigma\pi}$) for the carbon atoms that consist of five-membered rings were 90.7(6)° and 91.0(5)°, respectively (Supplementary Tables 4 and 6). These values are markedly smaller than those of C₆₀ (101.6°) and corannulene (98.2°), and comparable to that of the ideal sp² carbon atom like in graphite (90.0°).⁴²

(Supplementary Information)

Supplementary Table 3: Selected Bond Angles and the Sum of Them around Each Carbon Atom in Five-Membered Rings of **3a**^a

Atom	Angle 1 / deg	Angle 2 / deg	Angle 3 / deg	Sum / deg	$\theta_{\sigma\pi}$ ^b
C1	C2–C1–C9 105.01(14)	C10–C1–C9 128.70(15)	C10–C1–C2 126.26(15)	359.97(25)	90.57
C3	C2–C3–C4 108.23(14)	C4–C3–C19 126.02(15)	C2–C3–C19 125.76(16)	360.01(26)	– ^c
C4	C9–C4–C3 108.31(14)	C5–C4–C3 130.33(16)	C5–C4–C9 121.08(16)	359.72(27)	91.74
C9	C4–C9–C1 107.22(15)	C8–C9–C4 118.84(15)	C8–C9–C1 133.83(15)	359.89(26)	91.08
C10	C11–C10–C18 105.28(13)	C1–C10–C11 125.55(15)	C1–C10–C18 129.15(15)	359.98(25)	90.46
C12	C11–C12–C13 108.39(15)	C11–C12–C25 125.18(16)	C13–C12–C25 126.41(15)	359.98(27)	90.47
C13	C18–C13–C12 108.07(14)	C14–C13–C18 121.26(16)	C14–C13–C12 130.55(15)	359.88(26)	91.14
C18	C13–C18–C10 107.23(15)	C17–C18–C13 118.88(16)	C17–C18–C10 133.84(15)	359.95(27)	90.73
				Average	359.92(9) 90.88(47)
C31	C32–C31–C39 104.60(14)	C40–C31–C39 128.92(15)	C40–C31–C32 126.47(15)	359.99(25)	90.33
C33	C32–C33–C34 107.93(14)	C49–C33–C34 126.11(15)	C32–C33–C49 125.96(16)	360.00(26)	90.00
C34	C39–C34–C33 108.06(14)	C35–C34–C33 130.83(16)	C35–C34–C39 121.09(16)	359.98(27)	90.46

C39	C31–C39–C34	107.78(15)	C34–C39–C38	118.81(16)	C31–C39–C38	133.41(15)	360.00(27)	90.00
C40	C41–C40–C48	104.92(14)	C31–C40–C41	126.16(15)	C31–C40–C48	128.91(15)	359.99(25)	90.33
C42	C41–C42–C43	107.90(15)	C41–C42–C55	125.01(17)	C43–C42–C45	127.03(15)	359.94(27)	90.81
C43	C42–C43–C48	108.11(14)	C44–C43–C48	120.95(17)	C42–C43–C44	130.93(16)	359.99(27)	90.33
C48	C43–C48–C40	107.47(15)	C43–C48–C47	118.92(16)	C40–C48–C47	133.60(15)	359.99(27)	90.33
Average							359.99(9)	90.32(26)

^a Data from X-ray crystal structure. Carbon atoms with C–H bonds are excluded because of the difficulty in identifying the geometry of carbon atoms without the assumption of the position of hydrogen atoms using AFIX program. ^b A σ – π interorbital angle estimated by use of the π orbital axis vector (POAV) analysis as shown in references S6 and S7. ^c The q_{sp} value could not be calculated because the sum of bond angles was slightly over 360° .

Supplementary Table 4: Selected Bond Angles and the Sum of Them around Each Carbon Atom in Five-Membered Rings of **3b**^a

Atom	Angle 1 / deg	Angle 2 / deg	Angle 3 / deg	Sum / deg	$\theta_{\sigma\pi}$ ^b	
C1	C2–C1–C9 105.35(15)	C10–C1–C9 128.36(16)	C10–C1–C2 126.28(17)	359.99(28)	90.33	
C3	C2–C3–C4 108.36(15)	C4–C3–C19 126.36(16)	C2–C3–C19 125.22(17)	359.94(28)	90.81	
C4	C9–C4–C3 107.59(16)	C5–C4–C3 131.27(17)	C5–C4–C9 120.97(17)	359.83(29)	91.35	
C9	C4–C9–C1 107.64(15)	C8–C9–C4 119.44(17)	C8–C9–C1 132.59(17)	359.67(28)	91.88	
C10	C11–C10–C18 105.39(15)	C1–C10–C11 126.21(17)	C1–C10–C18 128.34(16)	359.94(28)	90.80	
C12	C11–C12–C13 106.91(15)	C11–C12–C12* 126.45(21)	C12*–C12–C13 126.63(20)	359.99(33)	90.33	
C13	C18–C13–C12 107.62(15)	C14–C13–C18 119.43(16)	C14–C13–C12 132.92(16)	359.97(27)	90.57	
C18	C13–C18–C10 108.12(15)	C17–C18–C13 120.51(16)	C17–C18–C10 131.11(17)	359.74(28)	91.67	
Average					359.88(10)	90.74(60)

^a Data from X-ray crystal structure. Carbon atoms with C–H bonds are excluded because of the difficulty in identifying the geometry of carbon atoms without the assumption of the position of hydrogen atoms using AFIX program. ^b A σ – π interorbital angle estimated by use of the π orbital axis vector (POAV) analysis as shown in references S6 and S7.

Supplementary Table 2: Selected Bond Angles and the Sum of Them around Each Carbon Atom in Five-Membered Rings of **4a**^a

Atom	Angle 1 / deg	Angle 2 / deg	Angle 3 / deg	Sum / deg	$\theta_{\sigma\pi}$ ^b
C1	C2–C1–C9 105.37(13)	C9–C1–C10 128.84(15)	C2–C1–C10 125.63(15)	359.84(25)	91.31
C3	C2–C3–C4 108.30(14)	C4–C3–C19 126.53(14)	C2–C3–C19 122.92(15)	359.75(25)	91.64
C4	C9–C4–C3 107.81(14)	C3–C4–C5 131.29(15)	C5–C4–C9 120.72(15)	359.82(25)	91.39
C9	C4–C9–C1 107.40(14)	C4–C9–C8 120.01(15)	C1–C9–C8 132.32(15)	359.73(25)	91.70
C10	C11–C10–C18 105.66(13)	C1–C10–C11 126.45(15)	C1–C10–C18 127.87(15)	359.98(25)	90.46
C12	C11–C12–C13 108.22(14)	C11–C12–C31 125.62(15)	C13–C12–C31 126.14(14)	359.98(25)	90.47
C13	C12–C13–C18 108.22(14)	C14–C13–C18 121.03(15)	C12–C13–C14 130.66(15)	359.91(25)	90.98
C18	C10–C18–C18 107.23(14)	C13–C18–C17 119.37(15)	C10–C18–C17 133.39(15)	359.99(25)	90.33
Average				359.88(9)	91.03(56)

^a Data from X-ray crystal structure. Carbon atoms with C–H bonds are excluded because of the difficulty in identifying the geometry of carbon atoms without the assumption of the position of hydrogen atoms using AFIX program. ^b A σ – π interorbital angle estimated by use of the π orbital axis vector (POAV) analysis as shown in references S6 and S7.

Supplementary Table 6: Selected Bond Angles and the Sum of Them around Each Carbon Atom in Five-Membered Rings of **4b**^a

Atom	Angle 1 / deg	Angle 2 / deg	Angle 3 / deg	Sum / deg	$\theta_{\sigma\pi}$ ^b
C1	C2–C1–C9 105.66(42)	C9–C1–C10 128.25(45)	C2–C1–C10 126.09(47)	360.00(77)	90.00
C3	C2–C3–C4 108.11(42)	C4–C3–C19 127.17(44)	C2–C3–C19 124.72(44)	360.00(75)	90.00
C4	C9–C4–C3 107.23(43)	C5–C4–C3 132.30(45)	C5–C4–C9 120.26(45)	359.79(77)	91.50
C9	C4–C9–C1 107.83(41)	C8–C9–C4 118.95(50)	C1–C9–C8 133.02(49)	359.80(81)	91.46
C10	C11–C10–C18 105.76(42)	C1–C10–C11 126.16(47)	C1–C10–C18 128.08(43)	360.00(76)	90.00
C12	C11–C12–C13 108.51(42)	C11–C12–C12* 126.05(58)	C12*–C12–C13 125.42(58)	359.98(92)	90.47

C13	C12–C13–C18	107.69(45)	C14–C13–C18	120.04(47)	C12–C13–C14	132.12(46)	359.85(80)	91.27
C18	C10–C18–C13	107.14(42)	C13–C18–C17	119.33(48)	C10–C18–C17	133.20(45)	359.67(78)	91.87
Average							359.89(28)	90.82(79)

^a Data from X-ray crystal structure. Carbon atoms with C–H bonds are excluded because of the difficulty in identifying the geometry of carbon atoms without the assumption of the position of hydrogen atoms using AFIX program. ^b A σ – π interorbital angle estimated by use of the π orbital axis vector (POAV) analysis as shown in references S6 and S7.

S6 Haddon, R. C. & Scott, L. T. π -Orbital conjugation and rehybridization in bridged annulenes and deformed molecules in general: π -orbital axis vector analysis. *Pure Appl. Chem.* **58**, 137–142 (1986).

S7 Haddon, R. C. GVB and POAV analysis of rehybridization and π -orbital misalignment in non-planar conjugated systems. *Chem. Phys. Lett.* **125**, 231–234 (1986).

[Comment #2-2]

Additionally, distances between oligomers should be provided to see if there is evidence of pi-pi interactions, and the authors should comment on the influence of the end-caps, phenyl or TMS-phenyl, in the molecular packing.

Reply & Revisions: We added the discussion on the influence of end-caps on the molecular packing based on the values of the interatomic C···C distance between stacked molecules in **3b** and **4b**. Since we have not obtained any experimental results of the physical properties that reflect this difference in intermolecular packing, we refrained from emphasizing the effect of end-caps on the intermolecular packing too much, yet focused on more fundamental aspects including the geometries (extent of pyramidalization) of each carbon atom. The paragraph now reads:

(page 6, 13th line in the first paragraph)

... Consequently, the π -conjugated chains of **3b** and **4b** exhibited substantially different S-shaped conformations with a C_i symmetry center and an arch-like shape with a C_2 symmetry axis, respectively. Accordingly, **3b** and **4b** adopted different packing motifs of the offset π -stacked arrays (Fig. 3c) and one-dimensional π -stacked columns (Fig. 3d), respectively. Reflecting the difference in packing motifs, the shortest interatomic C···C distance between stacked molecules in **3b** of 3.372 Å was significantly smaller than that in arched-shape **4b** (3.654 Å) and even slightly smaller than the sum of the van der Waals radii of the carbons, 3.40 Å, indicative of more effective intermolecular interaction of π -orbitals.

Despite the marked difference in the conformation and the packing motifs, the torsional angles in **3b** and **4b** were sufficiently small for extending π -conjugation over the main chains. ...

Along with the above-mentioned revision, the values of the closest intermolecular C···C distances between adjacent molecules were added into Fig. 3.

Fig. 3. X-ray crystal structures of biindenylidene dimers **3b** (a, c) and **4b** (b, d) drawn by thermal ellipsoid plots (50% probability for thermal ellipsoids): gray, carbon; white, hydrogen; yellow, silicon; green, chlorine. **a, b.** Top view of **3b** and **4b**. CH_2Cl_2 molecules in the crystal lattices of **4b** are omitted for clarity. **c, d.** Crystal-packing structure of **3b** and **4b**. The shortest intermolecular $\text{C}\cdots\text{C}$ distances were shown.

[Comment #3-1]

The electrochemistry and the optical absorption for the biindenylidene oligomers is fascinating. However, the manuscript would significantly benefit from the structure-property relationships being succinctly summarized as was done for fullerene in Fig. 1a. More importantly, the cyclic-voltammetry measurements for **3a-c** should also be provided. If the compounds are solubility limited, then authors should test the solubility limits in a wide variety of solvents and comment on their findings in the manuscript. The authors should also provide the energy levels (HOMO/LUMO) versus vacuum, which can be easily calculated using the estimates provided from CV and compare these to the calculated values in Fig. 5b.

Reply & Revisions: Thanks for bringing this up. We agree on the importance of including electrochemical data for **3a-c**, which have non-substituted phenyl groups at both ends. The electrochemical properties of **3a-c** in THF were evaluated, and their cyclic voltammograms were added in Supplementary Fig. 5. The electrochemical data were also summarized in Supplementary Table 7, which includes the LUMO energy levels versus vacuum.

We have selected THF as the solvent since it is well known to be effective in stabilizing anionic species, albeit with the low solubility of **3c** in THF (0.23 g L^{-1} ; equivalent to 0.28 mM). Accordingly, the fourth redox wave and beyond in **3c** were unfortunately masked by the reduction of THF, resulting in uncertain data on the reversibility of the fourth wave and whether the fifth wave was observed. However, these results are most likely due to the low current value for redox processes in **3c** caused by low solubility, and we believe that these do not affect the original conclusion. Notably, the redox waves in **3a-c** are slightly shifted to the negative direction compared to those in silyl-substituted **4a-c**, albeit with potential differences not exceeding ca. 0.1 V . These subtle energy shift indicates that the silyl groups at both ends have a negligible electronic effect. Given these results, we have revised the manuscript as follows:

(page 7)

Electrochemical properties. To corroborate the dependence of the redox properties on the number of five-membered rings, the electrochemical properties of oligo(biindenylidene)s **3a–c** and **4a–c** were examined using cyclic voltammetry (Fig. 4a, Supplementary Figs. 5 and 6, and Supplementary Table 7). The cyclic voltammograms of trialkylsilylphenyl-capped **4a**, **4b**, and **4c** in tetrahydrofuran (THF) showed two-, four-, and five-step reversible redox processes in the reductive region, respectively (Fig. 4a), and irreversible redox processes in the oxidative region (Supplementary Fig. 6). Considering that the first redox wave of **4c** was characterized by two-electron redox processes based on peak current analyses (Supplementary Discussion 2), **4a**, **4b**, and **4c** underwent two-, four-, and six-electron reductions, respectively, within the electrochemical window of THF. The phenyl-capped oligomers **3a–c** also showed cyclic voltammograms essentially similar to those of **4a–c**, except that **3c** showed only four-step redox processes in the reductive region within the electrochemical window of THF (Supplementary Fig. 5). The absence of the fifth redox wave might be attributable to the low solubility of **3c** in THF (0.23 g L^{-1}), resulting in the fifth redox wave with relatively low current values masked by the increase in baseline current due to the reduction of THF. Overall, these results demonstrate that oligo(biindenylidene)s can accept electrons equal to the number of five-membered rings in their main chains. ...

(Supplementary Information)

Supplementary Fig. 5: Cyclic voltammograms of **3a–c** measured at a scan rate of 0.1 V s^{-1} in tetrahydrofuran using $[n\text{-Bu}_4\text{N}][\text{PF}_6]$ (0.1 M) as the supporting electrolyte. All potentials are referenced against the ferrocene/ferrocenium (Fc/Fc^+) couple.

Supplementary Table 7: Electrochemical Data for Oligo(biindenylidene)s **3a–c** and **4a–c** in THF

Cmp d	$E_{1/2,\text{red}1} /$ $\text{V}^{a,b}$	$E_{1/2,\text{red}2} / \text{V}$ a,b	$E_{1/2,\text{red}3} / \text{V}$ a,b	$E_{1/2,\text{red}4} / \text{V}$ a,b	$E_{1/2,\text{red}5} / \text{V}$ a,b	LUMO / eV^d
	$(E_{\text{pc}1} / \text{V}^{a,c})$	$(E_{\text{pc}2} / \text{V}^{a,c})$	$(E_{\text{pc}3} / \text{V}^{a,c})$	$(E_{\text{pc}4} / \text{V}^{a,c})$	$(E_{\text{pc}5} / \text{V}^{a,c})$	
3a	-1.51 (-1.60)	-1.84 (-1.93)	n.d. ^e	n.d. ^e	n.d. ^e	-3.68
3b	-1.20	-1.36	-2.26	-2.75	n.d. ^e	-3.97

	(-1.32)	(-1.48)	(-2.38)	(-2.90)		
3c	-1.13	-1.87	-2.10	-2.59 ^f	n.d. ^e	-4.07
	(-1.19)	(-1.92)	(-2.16)	(-2.71)		
4a	-1.48	-1.78	n.d. ^e	n.d. ^e	n.d. ^e	-3.70
	(-1.57)	(-1.87)				
4b	-1.20	-1.33	-2.22	-2.60	n.d. ^e	-3.97
	(-1.31)	(-1.44)	(-2.33)	(-2.79)		
4c	-1.09	-1.82	-2.03	-2.61	-2.74	-4.09
	(-1.15)	(-1.87)	(-2.07)	(-2.67)	(-2.90)	
C ₆₀	-0.89	-1.47	-2.03	-2.50	n.d. ^e	-4.28
	(-0.94)	(-1.50)	(-2.07)	(-2.55)		

^a The redox potential determined by cyclic voltammetry under the following conditions: sample (1 mM) and [Bu₄N][PF₆] (0.1 M) in THF; scan rate 0.1 V s⁻¹. The potential was calibrated relative to the ferrocene/ferrocenium (Fc/Fc⁺) couple. ^b Half-wave redox potential. ^c Peak cathodic potential. ^d Energy level based on the Fermi scale estimated using the values of the onset potentials of the cyclic voltammograms using the following equation, $E_{\text{LUMO}}/\text{eV} = -(E_{\text{onset,red}} + 5.1)$, described in ref. S8. ^e Not detected. ^f The value of half-peak potential $E_{\text{pc}/2}$, i.e., the potential at half the maximum current in the cyclic voltammogram as a way to approximately estimate $E_{1/2}$. This $E_{\text{pc}/2}$ value is shown because of the uncertainty of the $E_{1/2}$ value due to the masking by the high baseline currents caused by the reduction of solvents. The $E_{\text{pc}/2}$ value is calculated based on the method described in ref. S9.

S8 Cardona, C. M., Li, W., Kaifer, A. E., Stockdale, D. & Bazan, G. C. Electrochemical considerations for determining absolute frontier orbital energy levels of conjugated polymers for solar cell applications. *Adv. Mater.* **23**, 2367–71 (2011).

S9 Roth, H. G., Romero, N. A. & Nicewicz, D. A. Experimental and Calculated Electrochemical Potentials of Common Organic Molecules for Applications to Single-Electron Redox Chemistry. *Synlett* **27**, 714–723 (2015).

In addition, based on the cyclic voltammogram of **3c**, we avoided stating definitively that "oligo(biindenylidene)s accept electrons equal to the number of five-membered rings" and revised the manuscript as follows as a more reasonable statement.

(p.1, Abstract)

Electrochemical studies corroborated that oligo(biindenylidene)s can accept electrons up to equal to the number of five-membered rings in their main chains.

(p.13, Conclusion)

Electrochemical studies of oligo(biindenylidene)s **4** revealed that these oligo(biindenylidene)s can accept electrons up to equal to the number of five-membered rings in their main chains and experimentally corroborated that the five-membered ring substructures play a crucial role in attaining robustness toward multi-electron reduction.

[Comment #3-2]

For the optical absorption measurements, the authors should include the optical bandgaps in Table S2.

Reply & Revisions: The values of the optical bandgap are frequently used to discuss the electronic structure of conjugated polymers and/or solid-state materials. However, since the present discussion focuses on the small molecule systems in a dilute solution, we refrained

from including the optical bandgap in this study. Alternatively, the values of absorption energy at the absorption maximum wavelengths are added in Supplementary Table 8.

Supplementary Table 8: Summary of Photophysical Data of **3a–c** and **4a–c**^[a]

Cmpd	$\lambda_{\text{abs}} / \text{nm}^{[b]}$	$E_{\text{abs}} / \text{eV}^{[c]}$	$\varepsilon / \text{M}^{-1} \text{cm}^{-1[d]}$
3a	482	2.57	1.32×10^4
3b	607	2.04	2.93×10^4
3c	653	1.90	5.34×10^4
4a	492	2.52	1.54×10^4
4b	612	2.03	3.23×10^4
4c	660	1.88	5.76×10^4

[a] In CH₂Cl₂. [b] Absorption maximum wavelength of the longest wavelength absorption band. [c] Absorption energy at an absorption maximum. [d] Molar extinction coefficient at the absorption maximum wavelength of the longest wavelength absorption band.

[Comment #3-3]

It is mentioned that 3a-c and 4a-c did not display any fluorescence. Was phosphorescence detected? The authors should include these measurements and comment on their findings.

Reply & Revisions: Based on the reviewer's comment on the possibility of phosphorescence, photoluminescence measurements of oligo(biindenylidene)s were newly attempted under cryogenic conditions in a degassed solvent, but any noticeable luminescence was observed. Although further insights into the reason for the non-luminescent character could be obtained by ultrafast transient absorption spectroscopy, we have concluded that this is beyond the scope of the present paper. Based on these findings, we tried to include comments about this topic. The paragraph now reads as follows:

(page 10, 5th line)

Conversely, **3a-c** and **4a-c** were virtually non-luminescent in a degassed 2-methyltetrahydrofuran solution even at 77 K despite the allowed nature of the $S_0 \rightarrow S_1$ transitions.

The corresponding experimental details are added in SI. The new texts now read:
(Supplementary Information, page S20)

Photoluminescence Spectroscopy. Photoluminescence spectra of **3a-c** and **4a-c** were attempted to be measured with a Hamamatsu Photonics Quantaaurus-QY Plus calibrated integrating sphere system C13534-02 equipped with a high-power Xe lamp L13685-01, a near-infrared multi-channel detector C13684-01, and a quartz-made Dewar vessel, using dilute sample solutions in 2-methyl tetrahydrofuran. The solvent was purified by vacuum distillation over CaH_2 , and the resulting sample solutions were degassed by purging the argon gas stream for ca.10 min. However, **3a-c** and **4a-c** were virtually non-luminescent in a degassed 2-methyltetrahydrofuran solution at both an ambient temperature and 77 K.

[Comment #4]

A central feature of fullerenes are the isotropic electron transport properties, and for C60 it can also be easily vacuum deposited to form interfacial layers in various inorganic/organic electronic devices. Although these biindenylidene oligomers show multi-electron reduction the electron transport properties need to be demonstrated. The authors should provide the electron mobility (SCLC and/or OFET) for 3a-c/4a-c to demonstrate the impact and utility. From the crystal structures, it can be seen these compounds have a preferred orientation, and so charge-transport may be anisotropic and occur preferentially in the vertical or horizontal directions, which the authors should also comment on.

Reply & Revisions: This is a very important point raised by the referee, and we are thankful for the comment. To gain insights into the electron conductivity of oligo(biindenylidene)s, the electronic photoconduction in a microcrystalline state was newly examined using flash-photolysis time-resolved microwave conductivity (FP-TRMC) measurements for **3a-c** and **4c**. As a result, biindenylidene trimers **3c** turned out to have moderate electron conductivity with the electron mobility μ_e of $0.06 \text{ cm}^2 \text{ V}^{-1} \text{ cm}^{-1}$. Comparison of the photoconductivity transient of **3c** with that of **4c** with distinct packing structure indicated that this electron conduction is predominantly characterized to be an intra-chain process. These results demonstrate that the intramolecular electron transport in the present systems is negligible in contrast to

excellent electron transport in C₆₀, yet the long-chain oligo(biindenylidene)s serve as good electron-transporting molecular wires. Based on these results, we added a new section “Electron conductivity” to the manuscript. The new section now reads:

(page 11)

Mobility of electrons. To gain insights into the electron mobility of oligo(biindenylidene)s, the electronic photoconduction in a microcrystalline state was examined using flash-photolysis time-resolved microwave conductivity (FP-TRMC).⁴³ Photoconductivity transients were observed for **3a–c** upon electrodeless photocarrier injection by the excitation at 355 nm (Fig. 6a and Supplementary Fig. 15). For **3a** and **3b**, almost no transient conductivity is observed whenever the excitation intensity is increased, or the crystal orientation is optimised. In stark contrast, **3c** exhibited fast decay with the transient conductivity $\phi\Sigma\mu$ (ϕ : photogeneration efficiency of the charge carriers, $\Sigma\mu$: sum of the isotropic electron and hole mobility) more than one order of magnitude larger than those of **3a** and **3b** (Fig. 6a), indicating that trimer **3c** exhibits superior charge carrier conductivity compared to the shorter-chain oligomers. The major contribution to the photoconductivity was determined to be electrons, given that the transient absorption spectrum of **3c** is in good agreement with the absorption band of radical anion **3c⁻** generated by electrochemical reduction (Fig. 6b and Supplementary Fig. 16). The photoinjected charge carrier density of **3c** was estimated by monitoring the transient absorption at 800 nm with absorption coefficients ε of $7 \times 10^4 \text{ M}^{-1} \text{ cm}^{-1} \text{ charge}^{-1}$ upon excitation at 355 nm, where photocarrier efficiency ϕ of 2×10^{-3} and intrinsic intramolecular electron mobility μ_e of $0.06 \text{ cm}^2 \text{ V}^{-1} \text{ s}^{-1}$ were obtained (Fig. 6c). Notably, the transient conductivities of **3c** and **4c** are comparable to each other despite the distinct crystal packing structures and π - π stacking distances (Figs. 6d, 3c, and 3d), indicative of the predominant contribution of intramolecular electron transport for their electronic photoconduction. These results demonstrated a promising utility of long-chain oligo(biindenylidene)s as electron-transporting molecular wires, although inter-chain electron transport is negligible in the present system.

Fig. 6: Electronic photoconduction in a microcrystalline state evaluated by flash-photolysis time-resolved microwave conductivity (FP-TRMC) measurements. **a.** Kinetic traces of photoconductivity transients recorded for polycrystalline **3a** (blue), **3b** (orange), and **3c** (red) under excitation at 355 nm, $4.6 \times 10^{15} \text{ photons cm}^{-2}$. **b.** Comparison of a photoconductivity transient (green) and transient optical absorption (red) at 800 nm recorded for **3c**. Given the absorption coefficient of **3c⁻** of ca. $7 \times 10^4 \text{ M}^{-1} \text{ cm}^{-1}$, the values of ϕ and μ of **3c** are calculated

to be 2×10^{-3} and $0.06 \text{ cm}^2 \text{ V}^{-1} \text{ s}^{-1}$, respectively. **c.** Transient absorption spectra observed for polycrystalline **3c** under excitation at 355 nm, 3.6×10^{16} photons cm^{-2} . Spectra were recorded at 0–100 ns (red), 1–1.1 ms (orange), 2–2.1 ms (green), and 4.8–4.9 ms (blue) after pulse exposure. **d.** The photoconductivity transients recorded for **3c** (red) and **4c** (green) in the logarithmic scale, suggesting non-pseudo first-order recombination kinetics for the decay

Accordingly, the Conclusion section is also revised. The paragraph now reads:

In contrast to the similarity with fullerenes in terms of electron affinity, the one-dimensional π -conjugation in oligo(biindenylidene)s resulted in pronounced absorption covering the entire visible region, unlike the weak absorption of C_{60} attributable to the highly symmetrical structure. Furthermore, FP-TRMC measurements also indicated that the oligo(biindenylidene)s with longer chain lengths serve as promising electron-transporting molecular wires. The current results highlight the significance of the pentagonal substructure for attaining stability toward multi-electron reduction and provide a new strategy for the molecular design of electron-accepting π -conjugated hydrocarbons.

In line with the above-mentioned revisions, Prof. Shu Seki (Kyoto University) joined as one of the co-authors because of his contribution to FP-TRMC measurements.

Although these results are not convincing regarding the practical utility of the present systems for electron-transporting materials, the excellent electron-transporting property is not the sole criterion of the value of electron-accepting π -electron systems. We are of the opinion that the main line of this paper should remain with the fundamentals of whether oligo(biindenylidene)s indeed have high electron affinity and stability toward multi-electron reduction comparable to fullerenes or not, as the origin of the high electron affinity and stability toward the multi-electron reduction of fullerenes has been simply unclear. While further optimizations of the packing structures by modulation of substituents and/or chain lengths might improve the electron-transporting properties, we feel that it is beyond the scope of the present paper. We will investigate them and report elsewhere in the near future.

Responses and changes made to Reviewer #3

(changes in the manuscript are highlighted in pink)

We thank the reviewer for the positive comments and careful inspection of our manuscript.

[Comment #1]

Figure 5 caption: "Inset: magnified spectra..." should be singular "Inset: magnified spectrum..."

Revisions: The grammatical error has been corrected as pointed out.

[Comment #2]

Reference 26: The first author's name is Bronstein (not Cronstein).

Revisions: The authors are sorry that a typo occurred, which has been corrected as pointed out.

Responses and changes made to Reviewer #4

(changes in the manuscript are highlighted in blue)

We appreciate the constructive criticism from the reviewer. Our reply to the reviewer's comments and what has been revised are listed as follows.

[Comment #1]

The authors stress the results are obtained in the absence of "curvature". But the X-ray of 4b has a curvature, which is not insignificant.

Reply & Revisions: Thanks also to this reviewer for bringing this up.

Please see also Comment #2-1 by Reviewer 2 for similar comments. Based on these reviewers' comments, we rephrased all the statements "curvature" to "pyramidalization of carbon atoms" to discriminate the geometry of each sp^2 carbon atom from the conformation of a molecular scaffold. In addition, to better clarify the geometries of the carbon atoms in oligo(biindenylidene)s in a quantitative manner, the sum of bond angles and the σ - π interorbital angles ($\theta_{\sigma\pi}$) of the carbon atoms in five-membered rings were calculated for the crystal structures, and newly added in Supplementary Tables 3–6.

[Comment #2-1]

Do the computations reproduce the structure of at least 3a and b? By looking at the image of fig 4 it seems not. But, this should be clarified.

Reply & Revisions: We chose the PBE0 density functional that best reproduces the structural parameters after screening several different types of DFT by comparing the geometries optimized by calculations with those obtained by the crystallographic analyses. The benchmark results were newly added in Supplementary Fig. 8:

Supplementary Fig. 8: Benchmark results of DFT calculations for oligo(biindenylidenes) **4a'**–**c'** using various density functionals. **a.** Selected C–C bond lengths. **b.** Orbital energy levels of Kohn-

Sham HOMOs and LUMOs.

Given that the biindenylidene dimers **3b** and **4b**, and trimers **3c** and **4c** should exist as a mixture of various metastable conformers because of their freely rotatable inter-unit C–C single bonds, it seems to be not realistic to reproduce the properties of oligo(biindenylidene)s in solution by taking all of the possible conformers into account within the limited computational resources and time. For this reason, we focused on the most stable conformers in the present calculations to gain reasonable insights into the electronic structure and physical properties. In this perspective, see also the reply to Comment #1 by Reviewer 1.

In addition, as for the solid-state structure, the slight conformational difference between the crystal structures and the optimized geometries by DFT calculations should reflect intermolecular interactions in the solid state. We believe this should not be considered a major problem, since it is well known that the conformations in the solid state are quite different from those in the gas phase or solution due to packing forces, especially for the flexible molecules.

[Comment #2-2]

Do the computed structure varies significantly going from the neutra to the reductive species?

Reply & Revisions: We thank the Reviewer for this comment, please also see Comments #1 and #2 in a similar direction by Reviewer 2. As shown in Fig. 4, theoretical calculations suggest that the distribution of C–C bond lengths changes significantly upon reduction. This large structural change is reasonable considering that indene acquires aromaticity in the reduced species.

Other revisions

(changes in the manuscript are highlighted in gray)

In addition to the revisions based on the reviewers' comments, we fixed several typographical errors as follows.

p.1, Abstract

Moreover, ultraviolet/visible/near-infrared absorption spectroscopy revealed that oligo(biindenylidene)s exhibit significantly enhanced absorption covering the entire visible region relative to C₆₀.

("in relation to" was changed to "relative to" to reduce wordiness)

p.5

"nuclear magnetic resonance (NMR) and mass spectroscopy"

is rephrased to

"nuclear magnetic resonance (NMR) spectroscopy and mass spectrometry"

p.9

Similar to conventional π -conjugated oligomers, the increase in chain lengths of oligo(biindenylidene)s from **3a** to **3c** and from **4a** to **4c** resulted in a substantial redshift and an increase in the molar absorption coefficients (ϵ) of the longest-wavelength absorption bands. In particular, the longest-wavelength absorption band of phenyl-capped **3c** with an absorption maximum wavelength (λ_{max}) of 653 nm was substantially red-shifted by 171 nm (5400 cm^{-1}) compared to that of the corresponding **3a** ($\lambda_{\text{max}} = 482 \text{ nm}$),...

(hyphens were added)

Reviewers' Comments:

Reviewer #2:

Remarks to the Author:

In the extensively revised manuscript titled "One-dimensional fragments of fullerene C₆₀ that exhibit robustness toward multi-electron reduction and pronounced light absorption", the authors (M. Hayakawa et al.) have thoroughly addressed all initial comments and concerns. Notably, the synthetic optimization, more thorough characterization, and demonstration of electron transport have substantially increased the impact of this work. In its revised form, these findings should garner broad interest from the scientific community, and it is therefore suitable for publication in Nature Communications.

Reviewer #4:

Remarks to the Author:

The authors have revised the manuscript and address my points and most of those of the other reviewers. Despite my objections, it appears that the paper will be published in Nature Communications. However, I do strongly believe that the statement "To elucidate the role of the five-membered ring substructures without the influence of high symmetry and curved structure" is misleading. The crystal structures of these molecules have curvature, thus the authors should soften this statement.

We appreciate the reviewers for their careful reviews and constructive comments on our manuscript. Our point-to-point response to the comments and what has been revised are listed as follows.

Responses and changes made to Reviewer #4

(changes in the manuscript are highlighted in blue)

The authors appreciate Reviewer 4 for the positive comments and several constructive suggestions, which we have addressed as described below thoroughly.

[Comment #1]

The authors have revised the manuscript and address my points and most of those of the other reviewers. Despite my objections, it appears that the paper will be published in Nature Communications. However, I do strongly believe that the statements "To elucidate the role of the five-membered ring substructures without the influence of high symmetry and curved structure" is misleading. The crystal structures of these molecules have curvature, thus the authors should soften this statement.

Reply & Revisions: In light of this reviewer's comment, we have thoroughly rephrased the description of "curvature" to "pyramidalized carbon atoms" to clarify that this manuscript focuses on the geometry of each sp^2 carbon atom rather than the conformation of the molecular scaffold. The texts now read as follows:

Page 1, the last sentence

Several structural factors have been suggested, including high symmetry, pyramidalized carbon atoms, and five-membered ring substructures. To elucidate the role of the five-membered ring substructures without the influence of high symmetry and pyramidalized carbon atoms, we herein report the synthesis and electron-accepting properties of oligo(biindenylidene)s, a one-dimensional fragment of fullerene C_{60} .

Page 2, 13th line in the second paragraph

Although Brunetti and coworkers reported π -extended 9,9'-bifluorenylidene derivatives composed of a C_{60} substructure without having pyramidalized carbon atoms,³¹ their highly twisted structures impede the effective extension of the π -conjugation, limiting the contribution of five-membered rings. A fragment molecule of C_{60} with effective π -conjugation between the five-membered ring substructures without pyramidalized carbon atoms is necessary to clarify the role of five-membered rings on the high electron affinity and robustness toward multi-electron reduction of fullerenes.

Page 3, Fig. 1

The texts "Curved structure" in Fig. 1a and "removal of curvature" in Fig. 1b were rephrased to "Pyramidalized carbon atoms" and "Elimination of pyramidalization around carbon atoms," respectively. Note that the figures and its legends also include several formatting changes from the previous version.

Fig. 1. Origin of prominent electron-accepting character of fullerenes. a. Three proposed structural factors for the electron-accepting character of fullerenes: **(i) high symmetry that results in the triply-degenerated lowest and second-lowest unoccupied orbitals, (ii) pyramidalized carbon atoms that mitigate bond angle strain around the carbon atoms upon reduction, and (iii) five-membered ring substructures that can acquire aromatic character in a reduced state.** b. Our molecular design of π -conjugated polymer **2** based on the hoop-shaped substructure **1** of C_{60} . c. End-capped oligo(biindenylidene)s **3** and **4** examined in this study.

Page 7, the last sentence

Overall, these results demonstrate that oligo(biindenylidene)s can accept electrons equal to the number of five-membered rings in their main chains. Accordingly, the five-membered ring substructure ensures the prominent stability of the π -conjugated hydrocarbons toward multi-electron reduction, even without pyramidalized carbon atoms or electron-withdrawing substituents.

However, in the sentence below in page 2, we have retained the expression "curved structure" for its conceptual suitability in expressing the allure of the structural characteristics of fullerene and fragment molecules in previous research. Nevertheless, I believe this expression will not lead to misunderstandings for readers, as it is not directly related to the origin of the electron-accepting properties of the fullerene focused on in this study, nor to the structural characteristics of the compound examined in this research.

Page 2, line 10

This fact indicates that the curved structure of the fullerenes is of significant interest.